# Cargo sorting zones in the *trans*-Golgi network visualized by super-resolution confocal live imaging microscopy in plants

Yutaro Shimizu[1,2], Junpei Takagi [3], Emi Ito[4], Yoko Ito [1,5], Kazuo Ebine[6,7], Yamato Komatsu[2], Yumi Goto[8], Mayuko Sato [8], Kiminori Toyooka [8], Takashi Ueda [6,7], Kazuo Kurokawa[1], Tomohiro Uemura [4✉] & Akihiko Nakano [1✉]

The *trans*-Golgi network (TGN) has been known as a key platform to sort and transport proteins to their final destinations in post-Golgi membrane trafficking. However, how the TGN sorts proteins with different destinies still remains elusive. Here, we examined 3D localization and 4D dynamics of TGN-localized proteins of *Arabidopsis thaliana* that are involved in either secretory or vacuolar trafficking from the TGN, by a multicolor high-speed and high-resolution spinning-disk confocal microscopy approach that we developed. We demonstrate that TGN-localized proteins exhibit spatially and temporally distinct distribution. VAMP721 (R-SNARE), AP (adaptor protein complex)−1, and clathrin which are involved in secretory trafficking compose an exclusive subregion, whereas VAMP727 (R-SNARE) and AP-4 involved in vacuolar trafficking compose another subregion on the same TGN. Based on these findings, we propose that the single TGN has at least two subregions, or "zones", responsible for distinct cargo sorting: the secretory-trafficking zone and the vacuolar-trafficking zone.

[1] Live Cell Super-Resolution Imaging Research Team, RIKEN Center for Advanced Photonics, Wako, Saitama, Japan. [2] Department of Biological Sciences, Graduate School of Science, The University of Tokyo, Bunkyo-ku, Tokyo, Japan. [3] Faculty of Science and Engineering, Konan University, Kobe, Hyogo, Japan. [4] Graduate School of Humanities and Sciences, Ochanomizu University, Bunkyo-ku, Tokyo, Japan. [5] Laboratoire de Biogenèse Membranaire, UMR 5200, CNRS, Université de Bordeaux, Villenave d'Ornon, France. [6] Division of Cellular Dynamics, National Institute for Basic Biology, Okazaki, Aichi, Japan. [7] The Department of Basic Biology, SOKENDAI (The Graduate University for Advanced Studies), Okazaki, Aichi, Japan. [8] Technology Platform Division, Mass Spectrometry and Microscopy Unit, RIKEN Center for Sustainable Resource Science, Yokohama, Kanagawa, Japan. ✉email: uemura.tomohiro@ocha.ac.jp; nakano@riken.jp

In eukaryotic cells, protein transport between intracellular compartments such as the endoplasmic reticulum (ER), the Golgi apparatus, the *trans*-Golgi network (TGN), endosomes, vacuoles, and the plasma membrane (PM) by membrane trafficking is essential for maintaining proper organellar and cellular functions. Secretory and vacuolar proteins, which are synthesized in the ER, enter the Golgi apparatus at the *cis*-cisterna, pass through medial- and *trans*-cisternae, and then reach the TGN en route to their final destinations, the PM and the vacuole[1–3]. The TGN was originally proposed as a specialized compartment adjacent to the *trans*-side of the Golgi apparatus, and is now known to be responsible for sorting of cargo proteins to their distinct destinations[1–3]. The plant TGN is morphologically characterized as a tubulovesicular structure, which produces transport carriers such as clathrin-coated vesicles (CCVs) and secretory vesicles[4,5]. In plant cells, the TGN also functions as an early endosome, which receives endocytosed cargos from the PM[6]. Thus, the plant TGN is a central sorting hub for the secretory, vacuolar, and endocytic trafficking.

SNAREs (soluble *N*-ethylmaleimide-sensitive factor attachment protein receptors) are a protein family, which mediate specific membrane fusion events in membrane trafficking. SNAREs can be divided into two groups, Q-SNAREs and R-SNAREs[7,8]. Three Q-SNAREs (Qa-, Qb-, and Qc-SNAREs) and one R-SNARE localize on target and transport vesicle membranes, respectively, and form a *trans*-SNARE complex composed of cognate partners on the target compartment[7,8]. Specific localization and proper pairing of Q-SNAREs and R-SNARE ensure accurate delivery of transport carriers to their correct destinations[7–9]. SYNTAXIN OF PLANTS 41/42/43 (SYP41/42/43) and SYP61 are TGN-localized Qa-SNARE and Qc-SNARE, respectively[10,11]. As R-SNARE, *Arabidopsis thaliana* harbors nine VAMP7 members (AtVAMP711–713, 721, 722, 724–727) that function in post-Golgi membrane trafficking[8,12]. They display different subcellular localization patterns, even though they share high sequence similarity[8,12]. For example, VAMP721 mainly localizes to the TGN and the PM, mediating secretory trafficking[13–16], whereas VAMP727 mostly localizes to multi-vesicular endosomes (the prevacuolar compartment) and the vacuolar membrane, mediating vacuolar trafficking[17]. In addition, some population of VAMP727 also localizes to the TGN, suggesting that VAMP727 mediates the trafficking from the TGN to the vacuole via multivesicular endosomes[16].

Adaptor protein (AP) complexes are another key protein family of post-Golgi membrane trafficking. They recognize specific amino-acid motifs of cargo proteins and pack them into nascent transport vesicles[18–21]. AP complexes are evolutionary conserved hetero-tetramer complexes, consisting of two large subunits ($\beta_{1–5}$ and $\alpha/\gamma/\delta/\epsilon/\zeta$), a medium subunit ($\mu_{1–5}$), and a small subunit ($\sigma_{1–5}$)[18–21]. Among five types of AP complexes, AP-1–AP-5, at least AP-1 and AP-4 function at the TGN in *A. thaliana*[18–29]. AP-1 appears to be involved in multiple trafficking pathways from the TGN to the PM and the vacuole[22–28], whereas AP-4 is involved in trafficking from the TGN to the vacuole[29,30].

In spite of the knowledge about these key components involved in the trafficking at the TGN, little is known how the TGN provides venues for sorting of secretory and vacuolar cargos simultaneously. In order to address this question, we have visualized TGN-localized proteins, SNAREs, AP complexes, and clathrin in plant cells, and examined their detailed localization and dynamics at high spatiotemporal resolution. The super-resolution confocal live imaging microscopy (SCLIM) we developed[31,32] allows us simultaneous multicolor high-speed and high-resolution observations, which have revealed that VAMP721/AP-1/clathrin and VAMP727/AP-4 are located on different subregions of a single TGN. We propose to call the subregions composed of VAMP721/AP-1/clathrin and VAMP727/AP-4 the "secretory-trafficking zone" and the "vacuolar-trafficking zone", respectively. Furthermore, detailed 4D observations suggest that portions of the secretory-trafficking zone are released from the TGN in the form of clusters of vesicles/buds, containing SYP61, AP-1, clathrin, and VAMP721. Our results suggest that the TGN has functional zones specialized for distinct trafficking pathways.

## Results

**Different distributions of SNAREs and V-ATPase on the TGN.** SYP41/42/43 and SYP61 have been studied as well-established TGN markers[10,11]. In addition, a subtype of V-ATPase is also known to reside at the TGN[33]. VHAa1 is an integral membrane protein subunit of the TGN-resident V-ATPase, which is essential for pH homeostasis of the TGN lumen[34]. We compared the distribution of these proteins with fluorescent tags on the TGN by conventional confocal laser scanning microscopy (CLSM) and SCLIM. They showed punctate patterns in 2D (CLSM) and round or ellipsoid shapes in 3D (SCLIM) (Fig. 1a–f). GFP-SYP43, mRFP-SYP61, and VHAa1-mRFP almost completely colocalized with each other by conventional CLSM as well as the control colocalization between GFP-SYP61 and mRFP-SYP61 (Fig. 1a–c). By SCLIM, GFP-SYP43 and mRFP-SYP61 also showed good colocalization like the control GFP-SYP61 vs mRFP-SYP61 (Fig. 1d, e). However, GFP-SYP43 and VHAa1-mRFP were significantly segregated when observed by SCLIM (Fig. 1f). A quantitative analysis based on the SCLIM data was carried out by calculating the correlation between GFP and RFP signal intensities per voxel on the TGN. The results confirmed that the extent of colocalization was low for SYP43 and VHAa1 (Fig. 1g), indicating that TGN-localized proteins with different functions are not uniformly distributed in the same TGN. By comparing to the location of *trans*-Golgi cisternae marked with ST-iRFP (infra-red fluorescent protein-tagged *trans*-membrane domain of a rat sialyl transferase[35]), we examined whether SYP43 and VHAa1 show any biased localization along the proximal–distal axis of the TGN, but could not observe a clear tendency (Supplementary Fig. 1a).

**Putative secretory- and vacuolar-trafficking zones in the TGN defined by R-SNAREs.** R-SNAREs play a role in guiding vesicles to target membranes where cognate Q-SNAREs are present, and cycle between donor and target membranes[2,7,8]. R-SNAREs are sorted into vesicles together with non-SNARE cargo proteins in the donor compartment, although the mechanisms for interaction with coat proteins are different[36–38]. Among the R-SNAREs that reside at the TGN, we decided to examine the behaviors of VAMP721 and VAMP727. As described above, these proteins are known to travel different routes after they leave the TGN, to the PM and to the vacuole, respectively[13–17]. We established Arabidopsis plants expressing iRFP-VAMP721 (iRFP-tagged VAMP721), TagRFP-VAMP727 (TagRFP-tagged VAMP727), and GFP-SYP61, and performed three-color 3D observation by SCLIM. As shown in Fig. 1h, i, iRFP-VAMP721 and TagRFP-VAMP727 localized to distinct regions of the same TGN labeled with GFP-SYP61. Similar results were obtained in comparison to another TGN marker, VHAa1 (Supplementary Fig. 1b, c). Hereafter, we call these TGN subregions "zones". The clear segregation of VAMP721 and VAMP727 on the TGN suggests that the TGN has spatially distinct zones for "secretory trafficking" to the PM and for "vacuolar trafficking". We refer to these as "secretory trafficking zones" and "vacuolar-trafficking zones" from here on. As a control, we compared localizations of GFP-, mRFP-, and iRFP-tagged SYP61 (Supplementary Fig. 1d, e). They overlapped very well with each other, indicating different effects

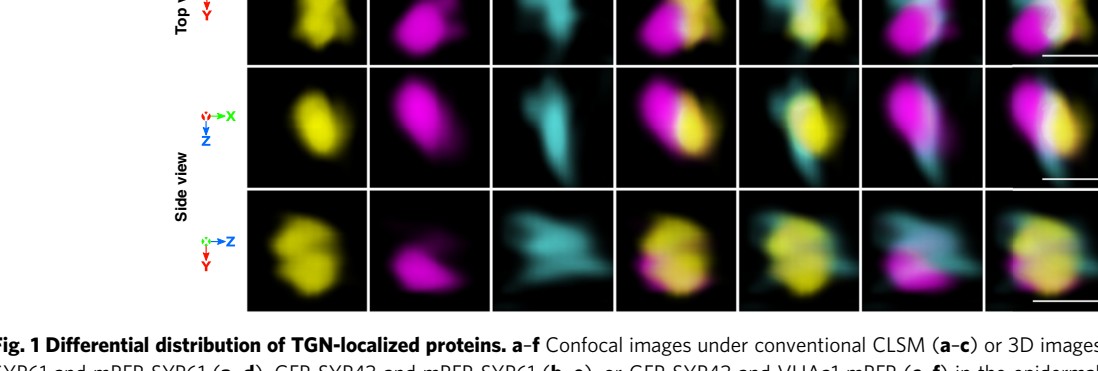

**Fig. 1 Differential distribution of TGN-localized proteins. a–f** Confocal images under conventional CLSM (**a–c**) or 3D images under SCLIM (**d–f**) of GFP-SYP61 and mRFP-SYP61 (**a**, **d**), GFP-SYP43 and mRFP-SYP61 (**b**, **e**), or GFP-SYP43 and VHAa1-mRFP (**c**, **f**) in the epidermal cells of the root elongation zone. **g** 3D colocalization analysis of the TGN markers: $n = 64$, 50, and 118 TGNs for GFP-SYP61 vs mRFP-SYP61, GFP-SYP43 vs mRFP-SYP61, and GFP-SYP43 vs VHAa1-mRFP, respectively, from five or more biological replicates. Two-sided Steel-Dwass test; $P = 0.86$ (Left: GFP-SYP61 × mRFP-SYP61 vs GFP-SYP43 × mRFP-SYP61), $P = 3.0 × 10^{-14}$ (Top: GFP-SYP61 × mRFP-YP61 vs GFP-SYP43 × VHAa1-mRFP), and $P = 2.2 × 10^{-7}$ (Right: GFP-SYP43 × mRFP-SYP61 vs GFP-SYP43 × VHAa1-mRFP); *$P < 0.01$, NS = nonsignificant. Boxes represent 25% and 75% quartiles, lines within the box represent the median, and whiskers represent the minimum and maximum values within 1.5× the interquartile range. **h** 3D images of GFP-SYP61, TagRFP-VAMP727, and iRFP-VAMP721 in the epidermal cell of the root elongation zone under SCLIM. **i** Multi-angle magnified 3D images of GFP-SYP61, TagRFP-VAMP727, and iRFP-VAMP721 of the boxed area in **h**. Upper panels: top view; middle and lower panels: side view (**h**, **i**). The experiments were repeated at least five times independently with similar results, and micrographs from representative experiments are presented. Scale bars = 5 μm (**a–c**); 1 μm (**h**, **i**). Grid width = 0.234 μm (**d–f**). Dashed lines indicate cell edges. V, vacuole area.

of different fluorescent tags or chromatic aberrations were not causing positional shifts of fluorescent signals.

**AP-1 and AP-4 are located in secretory- and vacuolar-trafficking zones in the TGN, respectively.** To investigate whether these trafficking zones labeled with the two R-SNAREs are

responsible for cargo sorting, we next focused our attention on the AP complexes AP-1 and AP-4. The plant AP-1 and AP-4 have been reported to colocalize with TGN markers; however, they do not colocalize with each other[24,29]. Thus, two possibilities are suggested: AP-1 and AP-4 localize on the different zones of the same TGN, or AP-1 and AP-4 are on the different TGNs.

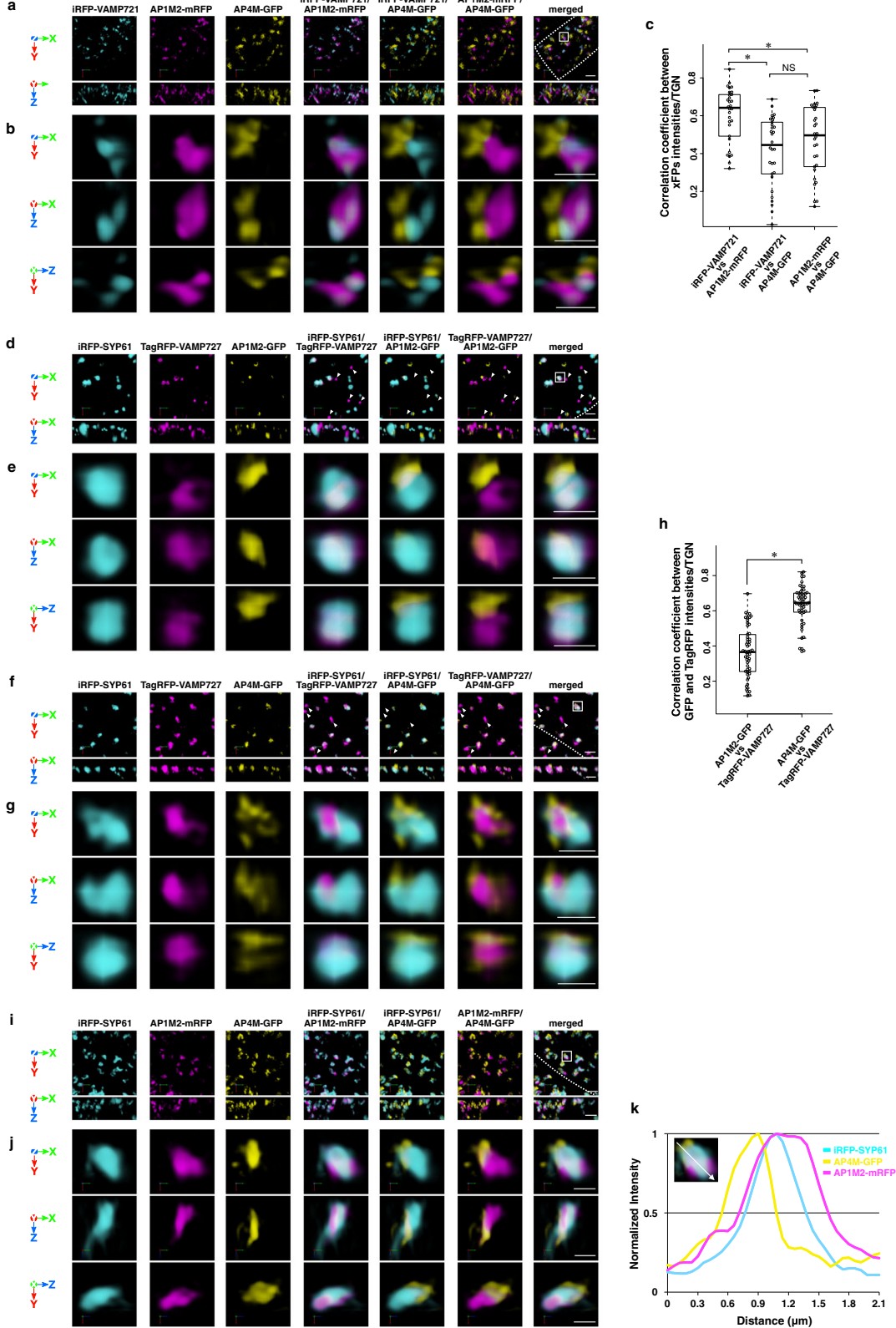

First, to investigate the involvement of AP-1 and AP-4 in secretory trafficking, we generated Arabidopsis plants expressing iRFP-VAMP721, AP1M2 (μ-subunit of AP-1)-mRFP, and AP4M (μ-subunit of AP-4)-GFP and performed SCLIM observation. As shown in Fig. 2a–c, AP4M-GFP did not colocalize with either AP1M2-mRFP or iRFP-VAMP721, whereas AP1M2-mRFP and iRFP-VAMP721 colocalized well. These results strongly suggest

that the AP-1 complex is in the secretory-trafficking zone of the TGN (Fig. 2a–c).

Next, to examine the roles of AP-1 and AP-4 in the vacuolar-trafficking zone, we generated Arabidopsis plants expressing either AP1M2-GFP or AP4M-GFP in addition to TagRFP-VAMP727 and iRFP-SYP61 and performed SCLIM observation. As shown in Fig. 2d–h, TagRFP-VAMP727 showed partial

**Fig. 2 Distinct suborganellar localization of VAMP721, VAMP727, AP-1, and AP-4. a–k** Three-color SCLIM imaging of root epidermal cells in the elongation zone of Arabidopsis expressing iRFP-VAMP721 × AP1M2-mRFP × AP4M-GFP (**a–c**), iRFP-SYP61 × TagRFP-VAMP727 × AP1M2-GFP (**d, e, h**), iRFP-SYP61 × TagRFP-VAMP727 × AP4M-GFP (**f–h**), or iRFP-SYP61 × AP1M2-mRFP × AP4M-GFP (**i–k**). **a, d, f, i** 3D images. **b, e, g, j** Multi-angle magnified 3D images of the boxed area in **a, d, f,** and **i**, respectively. Upper panels: top view; middle and lower panels: side view. Scale bars = 2 µm (**a, d, f, i**); 1 µm (**b, e, g, j**). Arrowheads indicate TagRFP-VAMP727 without iRFP-SYP61 and APs-GFP signals (**d, e**). Dashed lines indicate cell edges. **c** 3D colocalization analysis between µ-subunits of APs and VAMP721 on the TGN: $n = 30$ TGNs for each experiment, from three biological replicates. Two-sided Steel-Dwass test; $P = 1.8 \times 10^{-4}$ (Left: iRFP-VAMP721 × AP1M2-mRFP vs iRFP-VAMP721 × AP4M-GFP), $P = 8.3 \times 10^{-3}$ (Top: iRFP-VAMP721 × AP1M2-mRFP vs AP1M2-mRFP × AP4M-GFP), and $P = 0.51$ (Right: iRFP-VAMP721 × AP4M-GFP vs AP1M2-mRFP × AP4M-GFP); *$P < 0.01$, NS = nonsignificant. Boxes represent 25% and 75% quartiles, lines within the box represent the median, and whiskers represent the minimum and maximum values within 1.5× the interquartile range. **h** 3D colocalization analysis between µ-subunits of APs and VAMP727 on the TGN: $n = 54$ TGNs for each experiment, from five biological replicates. Two-sided Wilcoxon rank-sum test; $P = 1.5 \times 10^{-14}$; *$P < 0.01$. Boxes represent 25% and 75% quartiles, lines within the box represent the median, and whiskers represent the minimum and maximum values within 1.5× the interquartile range. **k** A graph shows normalized fluorescence intensity profile across a TGN of boxed area in **i**. The experiments were repeated independently three (**a–c**) or five (**d–k**) times with similar results, and photographs from representative experiments are presented.

colocalization with AP4M-GFP but not with AP1M2-GFP. It should be noted that TagRFP-VAMP727 without the iRFP-SYP61 signal, which probably represent VAMP727 in multivesicular endosomes, did not colocalize with either AP1M2-GFP or AP4M-GFP (Fig. 2d, f, arrowheads). Taken together, these results suggest that AP-4 is present in the vacuolar-trafficking zone of the TGN. In addition, AP1M2-mRFP and AP4M-GFP localized on distinct zones of the same TGN labeled with iRFP-SYP61 and/or FM4-64 (lipophilic dye) (Fig. 2i–k, Supplementary Movie 1, and Supplementary Fig. 1f–h).

**Clathrin is present in the secretory-trafficking zone of the TGN.** Clathrin is a coat protein to shape membranes to form CCVs, which are major carriers of cargo proteins at the PM and the TGN[21]. The AP complexes, AP-1 and AP-2, are known to interact with clathrin[20,39]. As compared to the well-characterized dynamics of clathrin/AP-2-mediated endocytosis[40–44], the behaviors of CCVs formed at the TGN remain largely unknown in plant cells. Our previous study by using conventional CLSM showed that fluorescent protein-tagged clathrin, CLC2-mKO (monomeric Kusabira Orange-tagged clathrin light chain 2) localized largely to the TGN, while its substantial portion was also found in the close proximity of the Golgi apparatus[45]. To gain an insight into more precise localization of clathrin in the Golgi/TGN region, we generated Arabidopsis plants expressing ST-iRFP, GFP-SYP61, and CLC2-mKO, and observed them by SCLIM. Consistent with our previous study[46], many TGNs labeled with GFP-SYP61 were found adjacent to the Golgi apparatus labeled with ST-iRFP (Fig. 3a–d). CLC2-mKO partially overlapped with GFP-SYP61 (TGN) and located on the opposite side to ST-iRFP (*trans*-Golgi), suggesting that clathrin accumulated on the distal side of the TGN membrane (Fig. 3a–d).

We next generated Arabidopsis plants expressing AP1M2-GFP or AP4M-GFP in addition to CLC2-mKO. SCLIM observation showed that AP1M2-GFP colocalized well with CLC2-mKO (Fig. 3e), but AP4M-GFP did not (Fig. 3f). The results of the correlation analysis are shown in Fig. 4g.

It should be noted that *A. thaliana* has three CLC isoforms (CLC1–3) and two clathrin heavy chain (CHC) isoforms (CHC1 and CHC2)[47]. Considering a possibility that different isoforms of clathrin may reside in different subpopulations of clathrin-coated structures, we compared the localization of fluorescent protein-tagged CLC1, CLC2, and CLC3 in Arabidopsis protoplasts. They showed very close localization to each other, indicating that their properties are similar at least for the subcellular localization (Supplementary Fig. 2a–c). We also observed the subcellular localization of endogenous CHCs by whole-mount immuno-fluorescence staining with the anti-CHC antibody that recognizes both CHC isoforms of *A. thaliana* (Agrisera, AS10 690). The

immunostaining images showed that endogenous CHCs colocalized well with CLC2-GFP and AP1M2-GFP but not with AP4M-GFP (Supplementary Fig. 2d–g). Taken together, these results suggest that clathrin mainly localizes to the secretory-trafficking zone of the TGN.

We next examined the protein interaction of AP-1 and AP-4 with clathrin by yeast two-hybrid and co-immunoprecipitation analyses. Since the large subunits of AP complexes have been shown to bind the amino-terminal domain of CHC[48,49], we compared the interaction between large subunits of AP-1 and AP-4 vs the amino-terminal domain of CHC2 by the yeast two-hybrid analysis. As shown in Fig. 3h, AP1G1 and AP1G2 (γ subunits of AP-1) but not AP1B1 or AP1B2 (β subunits of AP-1) interacted strongly with the amino-terminal domain of CHC2. The interaction of CHC2 amino-terminal domain was not detected with either AP4B (β subunit of AP-4) or AP4E (ε subunit of AP-4). We also investigated the interaction by the co-immunoprecipitation analysis using Arabidopsis plants expressing either AP1M2-GFP, AP4M-GFP, AP2M-GFP (positive control[44]), or free-GFP (negative control) with antibodies against GFP or CHCs. The results (Fig. 3i, j) showed that the anti-GFP brought down a significant amount of CHC when AP1M2-GFP was expressed but did much less when AP4M-GFP was expressed (see lanes no. 13 and 14). The biochemical evidence for the interaction of AP-1 with clathrin has been shown in a previous study[23], and our results are consistent with it. It is notable that a small amount of CHC was also detected in the anti-GFP immunoprecipitates from AP4M-GFP-expressing plants (quantification is shown in Fig. 3j). These results indicate that clathrin is indeed in the secretory-trafficking zone of the TGN represented by AP-1, although the involvement of a small portion of clathrin in the vacuolar-trafficking zone marked by AP-4 is not ruled out.

**Temporal dynamics of the secretory- and vacuolar-trafficking zones of the TGN.** Having revealed the spatial features of the TGN zones by 3D imaging, we next investigated their temporal dynamics by 4D imaging. First, Arabidopsis plants expressing iRFP-SYP61, AP1M2-mRFP, and AP4M-GFP as markers of the TGN, the secretory-trafficking zone, and the vacuolar-trafficking zone, respectively, were subjected to 4D imaging analysis by the simultaneous three-color, high-resolution, and high-speed SCLIM observation. As shown in Fig. 4 and Supplementary Movie 2, we frequently observed budding of the AP1M2-mRFP signal from the TGN without AP4M-GFP as structures around 500 nm in diameter (arrowheads). We also noticed that portions of iRFP-SYP61 left together with AP1M2-mRFP sometimes (Fig. 4b–d, f–h). The budding of AP4M-GFP from the TGN was not seen during our observations. From these results, we concluded that

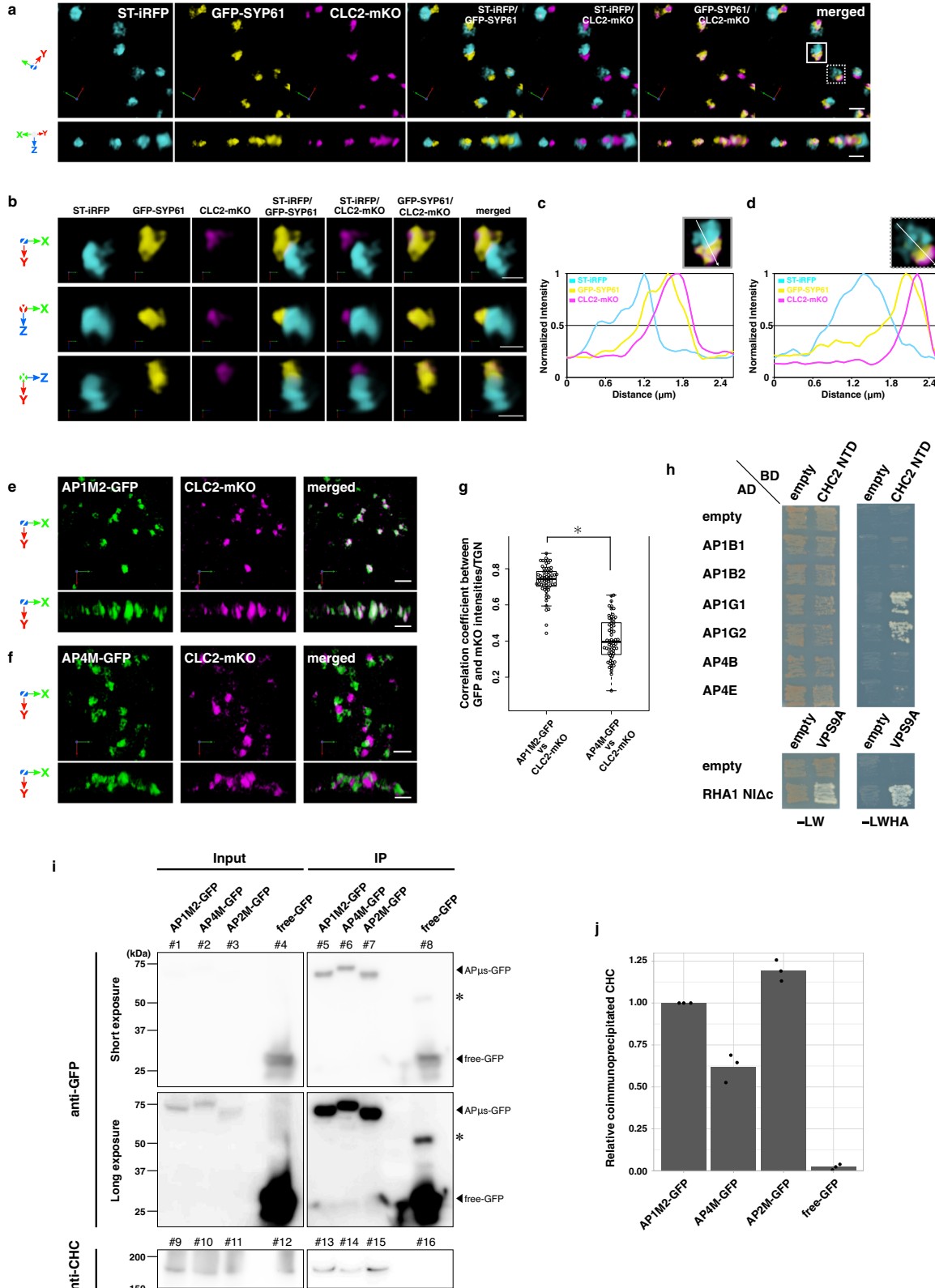

the zones for secretory trafficking and vacuolar trafficking are not only spatially distinct but have different dynamics.

Next, we examined the dynamics of AP-1, AP-4, and VAMP721 by the simultaneous SCLIM observation. Arabidopsis plants expressing iRFP-VAMP721, AP1M2-mRFP, and AP4M-GFP showed that AP1M2-mRFP and iRFP-VAMP721 behaved

together and simultaneously left the TGN labeled with the three-color fluorescence (Fig. 5a–d, arrowheads). On the other hand, AP4M-GFP behaved independently from both iRFP-VAMP721 and AP1M2-mRFP (Fig. 5a–d). These results suggest that the AP-1 and VAMP721 coordinately function in the same secretory-trafficking pathway. Furthermore, a small punctum containing

**Fig. 3 Clathrin localizes on the *trans*-side membrane of the TGN with AP-1. a** 3D images of ST-iRFP, GFP-SYP61, and CLC2-mKO in the epidermal cell of the root elongation zone under SCLIM. **b** Multi-angle magnified images of boxed area in **a**. **c**, **d** Graphs show normalized fluorescence intensity profile across a Golgi apparatus to clathrin of boxed area and dashed boxed area in **a**, respectively. **e**, **f** 3D images of AP1M2-GFP and CLC2-mKO (**e**) or AP4M-GFP and CLC2-mKO (**f**) in the epidermal cells of the root elongation zone under SCLIM. Upper panels: top view; middle and lower panels: side view (**a**, **b**, **e**, **f**). Scale bars = 2 μm (**a**, **e**, **f**); 1 μm (**b**). **g** 3D colocalization analysis between μ-subunits of APs and CLC2: $n = 60$ TGNs for each experiment, from five biological replicates. Two-sided Wilcoxon rank-sum test; $P = 2.2 \times 10^{-16}$; *$P < 0.01$. Boxes represent 25% and 75% quartiles, lines within the box represent the median, and whiskers represent the minimum and maximum values within 1.5× the interquartile range. **h** Yeast two-hybrid interaction assay between AP-1 or AP-4 vs clathrin. Large subunits of AP-1 and AP-4 were expressed as the fusion protein with an activation domain (AD) and an amino-terminal domain of CHC2 (CHC2 NTD) was expressed as the fusion protein with a DNA binding domain (BD) in the yeast strain AH109. Transformants were plated on medium lacking Leu, Trp, His, and adenine (-LWHA) to test for interactions between two proteins or on medium lacking Leu and Trp (-LW) for 4 days at 30 °C. The assay for RHA1 NIΔc vs VPS9A was performed as a control experiment. **i** Co-immunoprecipitation analysis of immunoprecipitates with an anti-GFP antibody from seedlings expressing either AP1M2-GFP, AP4M-GFP, AP2M-GFP (positive control), or free-GFP (negative control). The immunoprecipitates were immunoblotted using anti-GFP (lanes #1–8) or anti-CHC antibodies (lanes #9–16). Asterisks indicate non-specific bands. Input = 16% (anti-GFP; lanes #1–4); 1% (anti-CHC; lanes #9–12). **j** Densitometric quantification of CHC co-immunoprecipitated with GFP-tags. The experiments were repeated independently five (**a**–**h**) or three (**i**, **j**) times with similar results, and micrographs from representative experiments are presented.

AP1M2-mRFP and iRFP-VAMP721 (arrows) separated from the TGN-detached structure with the dual-color fluorescence (arrowheads) (Fig. 5a).

We further examined the temporal relationship between AP-1, AP-4, and clathrin. 4D imaging of AP1M2-GFP or AP4M-GFP and CLC2-mKO showed that CLC2-mKO behaved always together with AP1M2-mRFP (Fig. 5e, f and Supplementary Movie 3). In contrast, the movement of CLC2-mKO occurred independently from AP4M-GFP (Fig. 5g, h and Supplementary Movie 4). These observations support the idea that clathrin functions in the secretory-trafficking zone, but not in the vacuolar-trafficking zone.

**Portions of the secretory-trafficking zone become fragmented after budding from the TGN.** To investigate the temporal dynamics of clathrin in relation to the Golgi and the TGN, we performed 4D imaging for Arabidopsis plants expressing ST-iRFP, GFP-SYP61, and CLC2-mKO as markers for the Golgi, the TGN, and the secretory-trafficking zone, respectively. In Fig. 6a, the TGN labeled with GFP-SYP61 at time 0 s was adjacent to the Golgi apparatus labeled with ST-iRFP (Fig. 6a, 0 s). We and others have previously reported that the plant TGN shows two different forms, the Golgi-associated TGN (GA-TGN) and the Golgi-released independent TGN (GI-TGN; also called free TGN)[4–6,16,46]. The TGN of Fig. 6a, 0 s, was obviously the GA-TGN. A part of the TGN marked by GFP-SYP61 was then detached as a new GI-TGN (Figs. 6a, 6.4–25.6 s, arrowheads; and Supplementary Movie 5). Interestingly, CLC2-mKO separated from the GA-TGN together with the newly formed GI-TGN and stayed with GI-TGN for some period of time (Fig. 6a, arrowheads and squares; and Supplementary Movie 5). The size of this CLC2-mKO signal was ~500 nm in diameter, which was similar to that of the budding AP-1 signals from the TGN shown in Fig. 4. Such a large size is much bigger than that of CCVs, with an average diameter of 60 nm, reported by electron tomography in plant cells[50] and PM-localized CLC2-mKO signals equivalent to clathrin-coated pits (Supplementary Fig. 3). As shown in Fig. 6a (32.0 s and later)–c, the GFP-SYP61 and CLC2-mKO signals eventually split and fragmented into distinct small structures. We also investigated the ultrastructure of the TGN by transmission electron microscopy. Arabidopsis root cells were fixed by high-pressure freezing/freeze substitution, and ultrathin sections were prepared. As shown in Fig. 6d, the GI-TGN appeared to be composed of multiple vesicles/buds including clathrin-coated profiles and sometimes a cluster of CCVs was found near the Golgi/TGN (Fig. 6e). These results suggest that GI-TGN forms a structure containing a cluster of vesicles/buds, some of which

have clathrin profiles. After leaving the Golgi, this structure may well get fragmented into separate vesicles. We also noticed that the CLC2-mKO fluorescence often increased on the GA-TGN before budding, perhaps reflecting gradual accumulation of clathrin on the GA-TGN membrane in advance of the release of the GI-TGN/CCV cluster (Supplementary Fig. 4 and Supplementary Movie 6).

## Discussion

It is widely accepted that the TGN plays a pivotal role as the protein-sorting platform for post-Golgi trafficking. Proteins with a variety of destinations are sorted there from each other and then further delivered to their final locations. To understand the molecular mechanisms of how they find correct carriers and paths, cutting-edge live imaging is deemed a powerful approach. However, the style of presence of the TGN is complicated and differs among species. The TGN is often associated with the Golgi complex and thus used to be thought as a part of the Golgi, but emerging evidence indicates that the TGN can function independently of the Golgi. Plant cells have great advantages to study the functions and the dynamics of the TGN in this regard. First, considering the structure of the Golgi apparatus, plant cells show beautifully stacked structures of Golgi cisternae, which are separate as ministacks in the cytoplasm[4,5], unlike mammalian cells building a huge Golgi ribbons[51] and yeast *Saccharomyces cerevisiae* scattering unstacked cisternae[32,52]. Another important feature of plant cells is the clear evidence for the TGN that is not directly associated with the Golgi[4–6,16,46]. We have previously demonstrated that the TGN not associated with the Golgi (Golgi-released independent TGN or GI-TGN) forms from the TGN that is associated with the Golgi (Golgi-associated TGN or GA-TGN)[16,46]. Such separation of the TGN from the main body of the Golgi is also seen in mammalian cells and Drosophila[53] and, needless to say, in *S. cerevisiae*, as well[32,52].

In the present study, we decided to apply the high-performance live imaging microscopy SCLIM we developed to visualize the events during protein sorting at the TGN in plant cells. We chose Q-SNAREs SYP43 and SYP61 and a V-ATPase subunit VHAa1 as TGN markers, and examined the behaviors of R-SNAREs VAMP721 and VAMP727 as cargo proteins destined for secretory and vacuolar trafficking, respectively. Important players involved in sorting, adaptor complexes, AP-1 and AP-4, and clathrin, were also visualized and analyzed.

First, we demonstrated that SCLIM had indeed high space-resolution as compared to conventional CLSM. SCLIM observation showed that the TGN markers SYP43 and SYP61 colocalized well but had a slightly different localization from VHAa1. Such

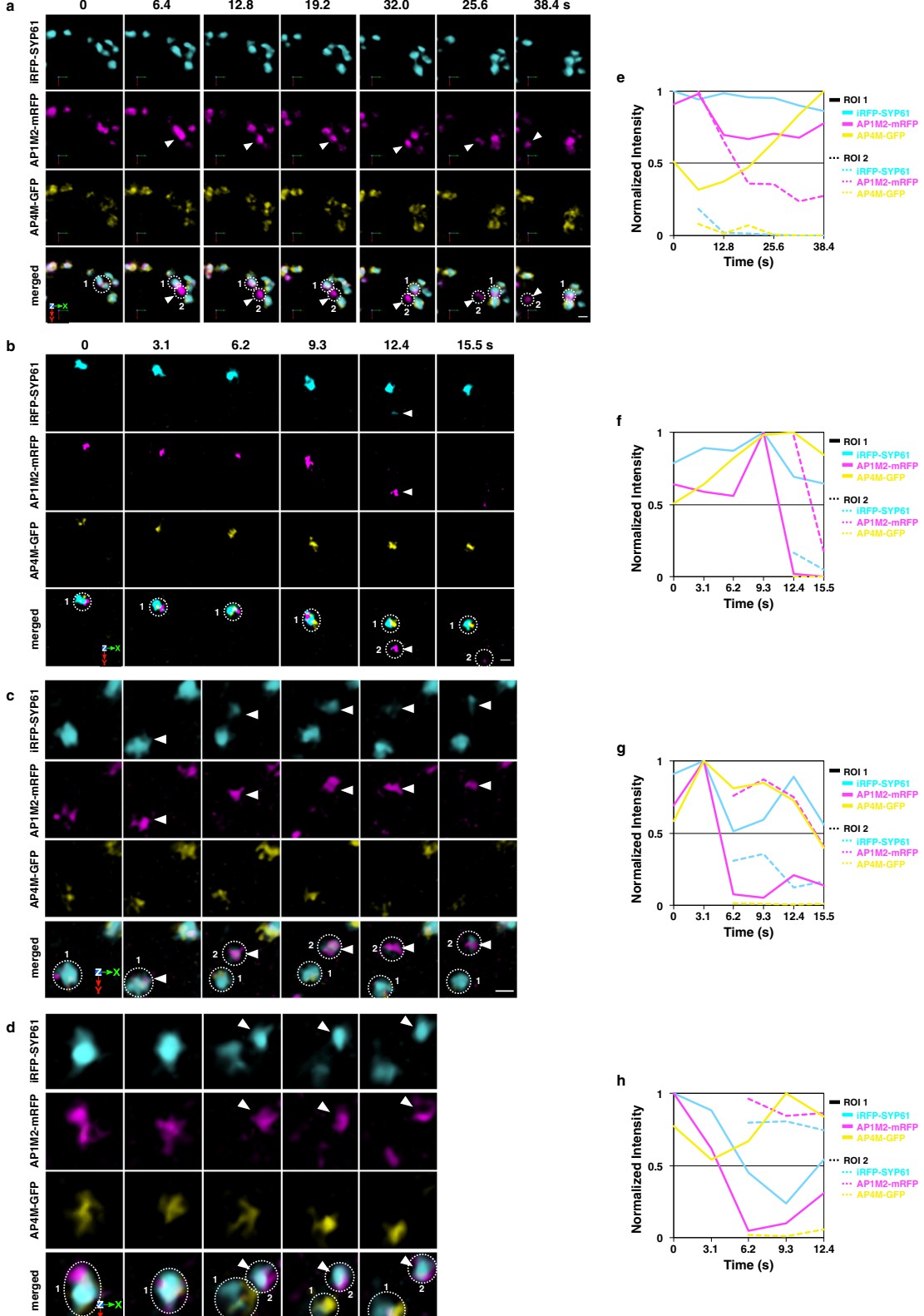

**Fig. 4 AP-1, but not AP-4, buds from the TGN. a–d** 3D time-lapse (4D) images of iRFP-SYP61, AP1M2-mRFP, and AP4M-GFP in the epidermal cells of the root elongation zone under SCLIM. Arrowheads indicate the dissociation of AP1M2-mRFP and iRFP-SYP61 from a large population of TGN labeled with iRFP-SYP61. Images are lined up every 6.4 s (**a**) or 3.1 s (**b–d**) from left to right. Scale bars = 1 μm. **e–h** Time course changes in relative fluorescence intensities of iRFP-SYP61, AP1M2-mRFP, and AP4M-GFP in ROIs of **a–d**. The experiments were repeated independently six times with similar results, and micrographs from representative experiments are presented.

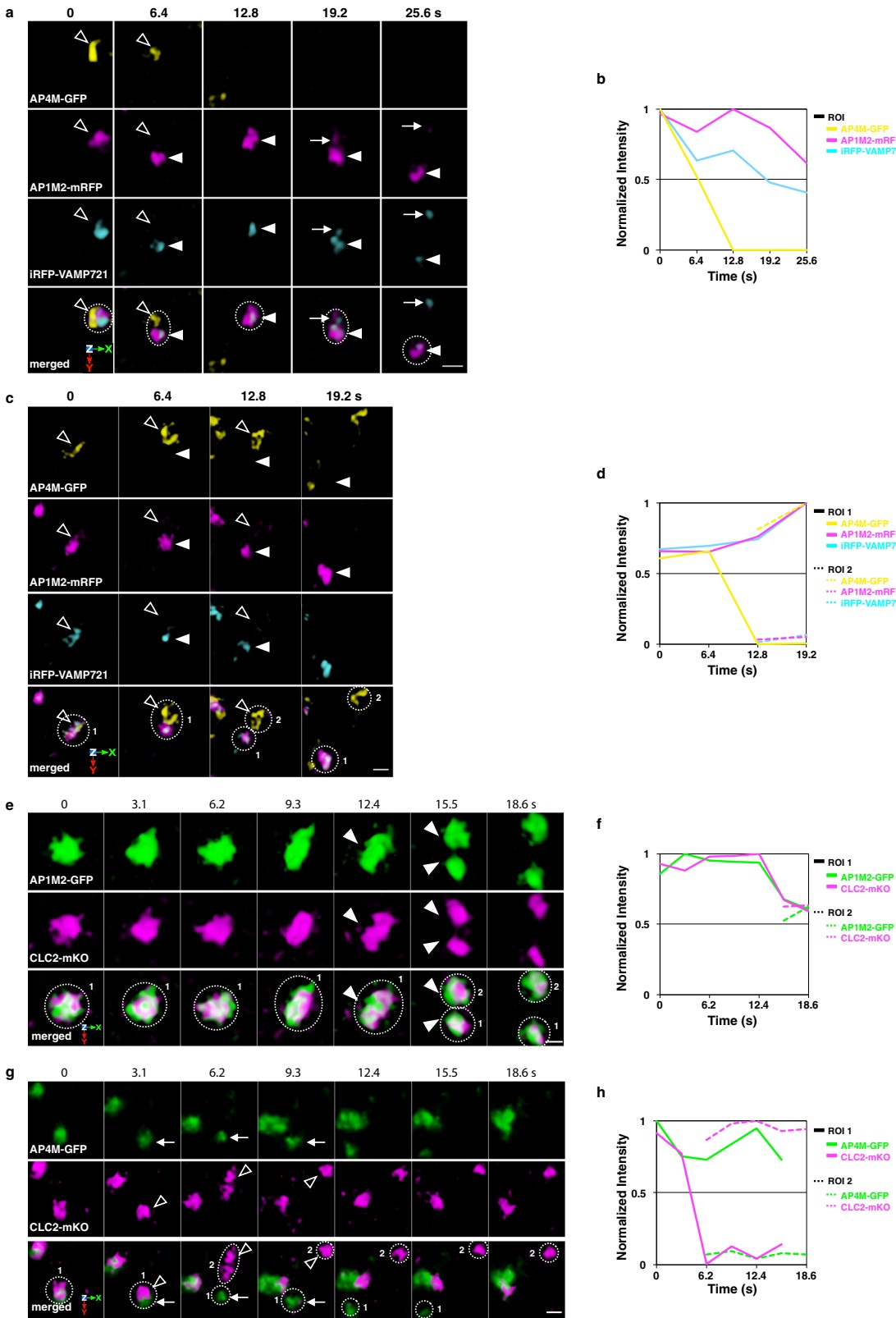

difference was not resolved by conventional CLSM (Fig. 1a–g). Next, we found that VAMP721 and VAMP727 showed clearly segregated localizations within the same TGN (Fig. 1h, i), suggesting that these localizations represent distinct zones in the TGN. Simultaneous observation of SYP61, VAMP721, and VAMP727 together with AP-1 and AP-4 indicated that VAMP721 vs AP-1 and VAMP727 vs AP-4 had high degrees of

colocalization as compared to other combinations. This finding supports the notion that AP-1 is involved in the sorting of VAMP721, which is destined for the PM, whereas AP-4 has a role in the sorting of VAMP727, which is to be transported to the vacuole (Fig. 2a–h). AP-1 and AP-4 were also clearly segregated from each other (Fig. 2i–k), again indicating that they represent distinct zones within the TGN, and this likely represents two

**Fig. 5 AP-1, but not AP-4, behaves together with VAMP721 and clathrin. a**, **c** 4D images of iRFP-VAMP721, AP1M2-mRFP, and AP4M-GFP in the epidermal cells of the root elongation zone under SCLIM. Arrowheads indicate AP1M2-mRFP and iRFP-VAMP721 dissociation from AP4M-GFP or the TGN (Open arrowheads). A small punctum containing AP1M2-mRFP and iRFP-VAMP721 (arrows) separated from the TGN-detached structure (arrowheads). **e**, **g** 4D images of AP1M2-GFP and CLC2-mKO (**e**) or AP4M-GFP and CLC2-mKO (**g**) in the epidermal cells of the root elongation zone under SCLIM. Arrowheads indicate coincidental AP1M2-GFP and CLC2-mKO signal fission. Open arrowheads indicate dissociation of CLC2-mKO from AP4M-GFP (arrows). Images are lined up every 6.4 s (**a**, **c**) or 3.1 s (**e**, **g**) from left to right. **b**, **d**, **f**, **h** Time course changes in relative fluorescence intensities in ROIs of **a**, **c**, **e**, and **g**, respectively. The experiments were repeated independently four times with similar results, and micrographs from representative experiments are presented.

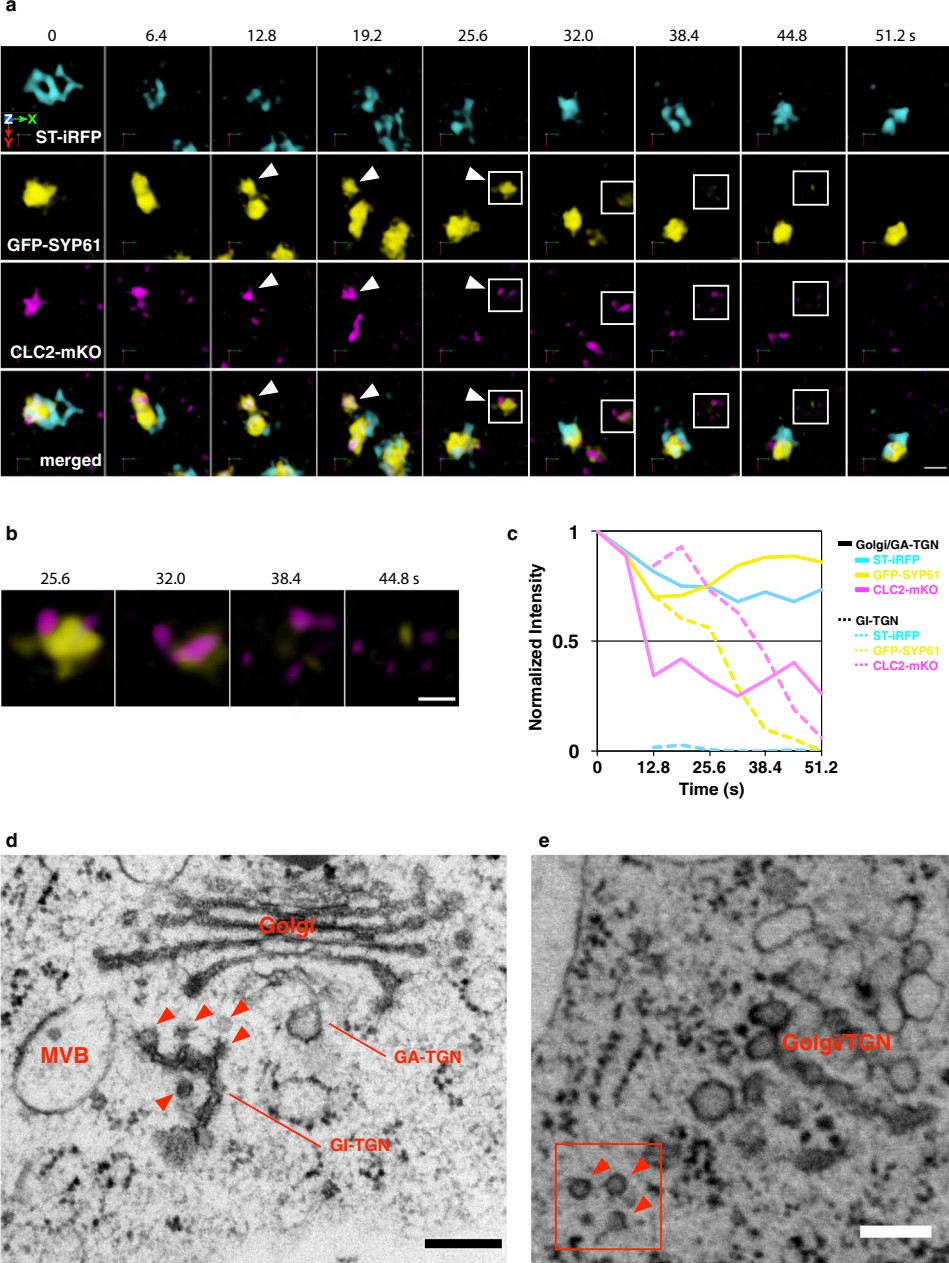

**Fig. 6 Temporal relation between and ultrastructure of the Golgi apparatus, TGN, and secretory trafficking zone component clathrin. a** 4D images of ST-iRFP, GFP-SYP61, and CLC2-mKO in the epidermal cell of the root elongation zone under SCLIM. White arrowheads indicate dissociation of CLC2 and SYP61 subpopulation from the major population of the TGN (GA-TGN) labeled with GFP-SYP61. **b** Magnified images of boxed area in **a**. Images are lined up every 6.4 s from left to right. **c** Time course changes in relative fluorescence intensities of ST-iRFP, GFP-SYP61, and CLC2-mKO in the Golgi/GA-TGN or GI-TGN area in **a**. **d**, **e** Transmission electron microscopic images of Arabidopsis root epidermal cells. Arrowheads indicate clathrin-coated vesicles/buds. Boxed area shows the CCV cluster. The experiments were repeated independently at least three times with similar results, and micrographs from representative experiments are presented. Scale bars = 1 μm (**a**); 500 nm (**b**); 200 nm (**d**, **e**).

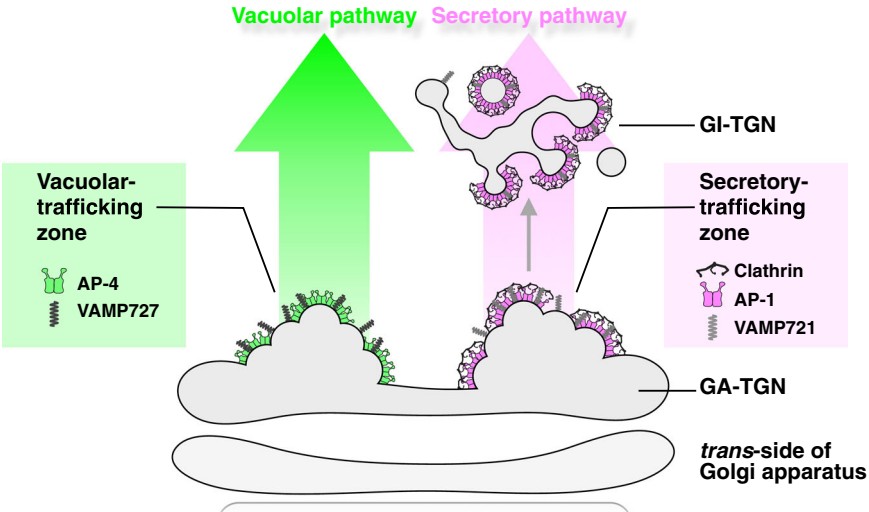

**Fig. 7 A schematic model of two distinct trafficking zones of the TGN.** The Golgi-associated TGN (GA-TGN) has at least two zones, the secretory-trafficking zone and the vacuolar-trafficking zone, which we propose are responsible for distinct cargo sorting in Arabidopsis root epidermal cells. The secretory-trafficking zone consists of R-SNARE VAMP721 (pale gray), adaptor protein AP-1 (magenta), coat protein clathrin (black). The vacuolar-trafficking zone consists of R-SNARE VAMP727 (dark gray) and adaptor protein AP-4 (green). From the secretory-trafficking zone, some of the TGN detaches as the Golgi-released independent TGN (GI-TGN) in a form of a vesicle/bud cluster including clathrin-coated vesicles/buds.

different trafficking paths (Fig. 7). Interestingly, clathrin, which was located to the distal side of the TGN showed very good colocalization with AP-1 but not with AP-4. This concept of sorting zones in the TGN is consistent with previous reports describing the low level of colocalization among TGN localized proteins[54–60]. It should be noted that the segregation of SYP43 and VHAa1 described above was not as large as that of AP-1 and AP-4 (compare Fig. 1f, g and Fig. 2c, i–k; median of colocalization correlation coefficient = 0.72 and 0.50, respectively). The reason for low colocalization of SYP43 and VHAa1 with overlapping residence is unknown at present.

Evolutionary diversification of APs is an interesting issue and has been discussed in several recent articles. Despite their structural conservation, however, functional similarities between different species are not well understood. Our present study provides clear visual demonstration that two of them, plant AP-1 and AP-4, localize to distinct zones in the TGN likely reflecting roles in secretory and vacuolar trafficking, respectively.

Several lines of evidence already exist to support the roles of VAMP721 and AP-1 in secretory trafficking[13–15,22,23,25]. VAMP721 (and its close paralog VAMP722) has been shown to form a PM SNARE complex with PEN1/SYP121 (Qa) and SNAP33 (Qb+Qc) and cytokinesis-specific SNARE complexes, one with KNOLLE/SYP111 (Qa) and SNAP33 (Qb/c) and another with KNOLLE/SYP111 (Qa), NPSN11 (Qb), and SYP71 (Qc), for secretory trafficking[13,14]. vamp721vamp722 double mutant seedlings exhibit defects in the secretory trafficking of PM-resident LTI6a/RCI2a and PIP2A/PIP2;1 (ref. [15]). ap1 mutants show defects in the secretory and recycling trafficking of invertase, secGFP, mucilage, BRASSINOSTEROID INSENSI-TIVE1, and auxin efflux carrier PIN2 to the PM[22,23,25]. vamp721vamp722 double mutant and ap1 mutant seedlings also exhibit incomplete cell plate formation and abnormal localization of Qa-SNARE KNOLLE/SYP111 (refs. [14,15,23,24]). The observations made in the present study, which showed not only the colocalization but also the synchronized dynamics of VAMP721 and AP-1, provide the first concrete image of how they cooperate during the sorting for the trafficking in a unique zone of the TGN. It should be noted here that the cooperation of VAMP721 and AP-1 as well as clathrin continues even after the TGN separates

from the Golgi to become GI-TGN. In the electron-microscopic observation shown in Fig. 6, the GI-TGN appears to contain multiple vesicles/buds, like the late/free-TGN[4,5] or the immature secretory vesicle clusters[61] as reported before, and may well be fragmented into separate vesicles at a later stage. The fate of individual vesicles after fragmentation will be an interesting topic, and we plan to pursue it by live imaging with higher spatio-temporal resolution in the future.

In contrast to the dynamic behavior of AP-1, AP-4 appears to persist on the GA-TGN for a longer period of time. The mechanism of AP-4-mediated protein transport remains elusive, but the mutant phenotype of AP-4 in sorting of vacuolar cargos with VACUOLAR SORTING RECEPTOR 1 indicates that AP-4 mediates vacuolar trafficking[29]. Unlike AP-1, AP-4 is not associated with clathrin in our SCLIM observations. Whether VAMP727 is released from the TGN in the form of vesicles or delivered to the vacuolar path by maturation-type or organelle-contact processes remains to be addressed. We have recently succeeded in the development of the second-generation SCLIM with a much higher spatiotemporal resolution and we are hoping to visualize these events in more detail in the near future. In addition, it will be important to establish a pulse-chase-type experimental system in order to monitor the passage of cargo proteins to examine how the observed TGN zones relate to different trafficking pathways.

## Methods
**Plant materials and growth conditions**. Surface-sterilized seeds were sown on Murashige and Skoog (MS) medium (1×MS salt, 2% sucrose, 1× Gamborg's vitamin mix, and 0.3% phytagel), vernalized at 4 °C for 2 days, and then grown at 23 °C under continuous light. The *A. thaliana* plants used in the present study were the ecotype Colombia-0 and transgenic lines expressing the following proteins: SYP61pro:GFP-SYP61, SYP61pro:mRFP-SYP61, SYP61pro:iRFP-SYP61, SYP43-pro:GFP-SYP43, VHAa1pro:VHAa1-GFP, VHAa1pro:VHAa1-mRFP, AP1M2pro:AP1M2-GFP, AP1M2pro:AP1M2-mRFP, AP2Mpro:AP2M-GFP, AP4Mpro:AP4M-GFP, VAMP721pro:iRFP-VAMP721, VAMP727pro:TagRFP-VAMP727, CLC2pro:CLC2-GFP, CLC2pro:CLC2-mKO, free-GFP, and UBQ10pro:ST-iRFP:NOSter. Original plants expressing GFP-SYP61, mRFP-SYP61, GFP-SYP43, TagRFP-VAMP727, CLC2-GFP, and CLC2-mKO were generated as reported previously[17,45,62,63]. The plants expressing VHAa1-GFP and VHAa1-mRFP were provided by Dr. Karin Schumacher (Heidelberg University, Germany). The plant expressing free-GFP was provided by Dr. Shoji Mano (National Institute for Basic

Biology, Japan). The plants expressing AP1M2-GFP, AP1M2-mRFP, AP2M-GFP, and AP4M-GFP were provided by Dr. Ikuko Hara-Nishimura (Konan University, Japan) and Dr. Tomoo Shimada (Kyoto University, Japan). The transgenic plants expressing combination of the fluorescent protein-tagged proteins of interest were generated by cross-pollination.

**Plasmid construction.** To generate SYP61pro:XFP-SYP61, the genomic DNA of *A. thaliana* SYP61 including approximately 2.3 kb of the 5′-upstream sequence and 0.8 kb of the 3′-downstream sequence was PCR-amplified and cloned into the pEN-TER/D-TOPO entry vector (Thermo Fisher Scientific). To generate VAMP721pro:iRFP-VAMP721, the genomic DNA of *A. thaliana* VAMP721 including approximately 2.0 kb of the 5′-upstream sequence and 0.9 of the kb 3′-downstream sequence was PCR-amplified and cloned as described above. The above clones were fused with a cDNA of GFP, mRFP, or iRFP713 (ref. [64]) using the In-Fusion HD Cloning Kit (Clontech) and then recombined into the destination vector pBGW or pHGW using the Gateway LR Clonase II enzyme mix (Thermo Fisher Scientific). To generate UBQ10pro:ST-iRFP:NOSter, the DNA corresponding with 52 N-terminal amino acids of a rat 2,6-sialyl transferase[35] was initially PCR-amplified and cloned into the entry vector containing 35S promoter, iRFP, and NOS terminator using In-Fusion HD Cloning Kit (Clontech). The promoter region of the entry vector containing 35Spro:ST:NOSter was then replaced with UBQ10 promoter using In-Fusion HD Cloning Kit (Clontech), and then recombined into the destination vector pBGW using the Gateway LR Clonase II enzyme mix (Thermo Fisher Scientific). For the protoplast transient expression assay, the PCR-amplified cDNA of CLC1, CLC2, or CLC3 was integrated in front of the cDNA encoding GFP or mRFP of binary vectors, modified pBluescript II KS (+), with the In-Fusion HD Cloning Kit (Clontech). To construct GAL4 AD-fusion vectors for the yeast two-hybrid analysis, the full-length coding sequences of large subunits of AP-1 and AP-4 (AP1/2B1: AT4G11380.1, AP1/2B2: AT4G23460, AP1G1: AT1G60070, AP1G2: AT1G23900, AP4B: AT5G11490, AP4E: AT1G31730) were subcloned into pAD-GAL4-GWRFC. To construct GAL4 BD-CHC2 NTD vector, the coding sequence for the amino-terminal domain of CHC2 (residues 1–536) was subcloned into SmaI-digested pBD-GAL4-Cam by the In-Fusion reaction. pAD-GAL4-GWRFC was provided by Dr. Taku Demura (Nara Institute of Science and Technology, Japan). BD-RHA1 NIΔc and AD-VPS9A have been reported previously[65]. The primers used in this study are listed in Supplementary Data 1.

**Fluorescence microscopy and image analyses.** We observed primary root epidermal cells in the elongation zone of the transgenic seedlings 7 days after germination (7 DAG). Conventional CLSM for 2D imaging was performed using a Zeiss LSM780 with a Plan-Apochromat 63×/NA 1.4 Oil objective lens, or an α Plan-Apochromat 100×/NA 1.57 Oil-Hi DIC Corr M27 objective lens as the high-resolution objective lens (Carl Zeiss). 3D and 4D imaging were performed with SCLIM[31,32]. The system is composed of an Olympus model IX-73 inverted fluorescence microscope with a UPlanSApo 100×/NA 1.4 Oil objective lens (Olympus), a high-speed spinning-disk confocal scanner (Yokogawa Electric), a custom-made spectroscopic unit, image intensifiers (Hamamatsu Photonics) with a custom-made cooling system, intermediate lenses (4× and 2/3×), and three EM-CCD cameras (Hamamatsu Photonics) for green, red, and infrared fluorescence channels. The pixel size corresponds to 0.06 μm on the sample plane. For 3D observations, we collected 51 optical sections spaced 0.1 μm apart (z-range = 5 μm). For 4D observations, we collected 21 or 31 optical sections spaced 0.1 μm apart, or 16 optical sections spaced 0.2 μm apart (z-range = 2–3 μm). The x and y axes spatial resolution of SCLIM is 180 nm, 180 nm, 240 nm for green, red, and infrared fluorescence channels, respectively. Z-stack images were reconstructed to 3D images and deconvolved by using theoretical point spread functions with Volocity (Quorum Technologies). Calculation of the Pearson correlation coefficients between the signal intensities of each voxel on the TGN was performed after three-dimensionally segmenting individual TGNs as ROI with Volocity. Thresholds for the calculation of correlation coefficients were automatically determined[66]. Signal intensity profiles of images were measured as previously reported with ImageJ[52].

To visualize the TGN with a lipophilic dye, Arabidopsis root cells were treated with 2 μM FM4-64 (Thermo Fisher Scientific). The image of FM4-64 was taken after 6 min of uptake. Fluorescence from GFP, RFP, and FM4-64 was unmixed using linear unmixing algorithms of ZEISS ZEN software.

Whole-mount immunolabelling of *Arabidopsis* roots with the CHC antibody was performed according to previous reports with minor modifications[58,67]. In brief, the seedlings 4 DAG were fixed with 4% paraformaldehyde in microtubule stabilizing buffer (MTSB: 50 mM PIPES, 5 mM EGTA, and 5 mM MgSO₄; pH 7.0 with KOH) for 1 h at room temperature (RT) and then washed four times with MTSB followed by twice with distilled water. Roots of the seedlings were cut on a polyethylenimine-coated glass (Iwaki), dried at RT, and rehydrated with MTSB. The roots were then permeabilized with 2% Driselase (Sigma) in MTSB for 35 min at RT, washed four times with MTSB, and treated with 10% dimethylsulfoxide + 3% Nonidet P-40 substitute (Nacalai Tesque) in MTSB for 1 h at RT. Nonspecific binding sites were blocked with 5% normal donkey serum (NDS; Jackson Immunoresearch) in MTSB for 1 h at RT. After blocking, the samples were incubated with anti-CHC primary antibody (Agrisera, AS10 690; 1:300 dilution) in 5% NDS/MTSB overnight at 4 °C and then washed four times with MTSB.

Subsequently, the samples were stained with AlexaFluor 594-conjugated donkey anti-rabbit IgG secondary antibody (Jackson Immunoresearch, 711-585-152; 1:300 dilution) in 5% NDS/MTSB for 1 h at RT and then washed four times with MTSB.

The protoplast transient expression assay of CLCs was performed by the method described previously with minor modifications[12,68]. Approximately 2 g of *Arabidopsis* suspension-cultured cells ("Deep" line) were incubated in 25 ml of polysaccharide-degrading enzyme solution [400 mM mannitol, 5 mM EGTA, 1% (w/v) cellulase Y-C, and 0.05% (w/v) pectolyase Y-23] for 90 min at 30 °C under gentle agitation and filtered with a nylon mesh (70-μm pore). Resulting protoplasts were collected by centrifugation at 250×g for 10 min, washed twice with 25 ml of washing solution (400 mM mannitol, 70 mM CaCl₂, and 5 mM MES-KOH 5.7), and resuspended in 1 ml of MaMg (400 mM mannitol, 15 mM MgCl₂, and 5 mM MES-KOH pH 5.7) at RT. After the addition of 20 μg of the plasmids and 50 μg of salmon sperm carrier DNA to 100 μl of protoplast solution, 400 μl of DNA uptake solution [400 mM mannitol, 40% (w/v) polyethylene glycol 6000, and 100 mM Ca(NO₃)₂] was added at RT. Subsequently, the protoplasts were incubated on ice for 20 min and then diluted with 10 ml of dilution solution (400 mM mannitol, 125 mM CaCl₂, 5 mM KCl, 5 mM glucose, and 1.5 mM MES-KOH pH 5.7) at RT. Resulting transformants were resuspended in 4 ml of MS liquid medium containing 400 mM mannitol and incubated with gentle agitation at 23 °C for 16 h in the dark. The transformed *Arabidopsis* protoplasts were observed with the Zeiss LSM780.

**Electron microscopy.** High-pressure freezing/freeze substitution and ultramicrotomy were performed as described previously[61]. In brief, Arabidopsis root cells (7 DAG) were frozen in a high-pressure freezer EM-ICE (Leica), substituted/fixed with acetone containing 2% osmium tetroxide. After washing with methanol, the samples were stained with 1% uranyl acetate in methanol and then embedded in epoxy resin. Ultrathin sections (80 nm) of the fixed sample were mounted on formvar-supported one-hole copper grids and stained with 4% uranyl acetate and lead citrate. The samples were acquired with a transmission electron microscope JEM-1400 Flash (JEOL).

**Yeast two-hybrid analysis.** The pAD and pBD vectors were transformed into *S. cerevisiae* strain AH109 (Clontech, Takara Bio) with a polyethylene glycol/lithium acetate protocol. The transformants with the both vectors were grown on solid synthetic dextrose medium lacking Leu and Trp. To examine interactions between the pAD- and pBD-fusion proteins, the transformants were grown on solid synthetic dextrose medium lacking Leu, Trp, His, and adenine for 4 days at 30 °C. AD-RHA1 NIΔc and BD-VPS9A were used for the control experiment. Empty vectors, pAD-GAL4-2.1 and pBD-GAL4-Cam, were used as negative controls. Five independent colonies were tested for each interaction.

**Coimmunoprecipitation analysis.** Coimmunoprecipitation analysis was performed according to the case of AP2M[44]. In brief, ~0.4 g of *A. thaliana*-expressing AP1M2-GFP, AP4M-GFP, AP2M-GFP, or free-GFP (14 DAG) were homogenized on ice in 1.2 ml of a lysis buffer (50 mM Tris-HCl, pH7.5, 1 mM EDTA, 1% Triton X-100, and a protease inhibitor cocktail in the absence of NaCl) and then centrifuged at 10,000×g for 10 min at 4 °C. The supernatants were immunoprecipitated with an anti-GFP antibody using μMACS GFP Isolation Kit (Miltenyi Biotec). Then, the immunoprecipitates were separated by SDS-PAGE and immunoblotted with an anti-GFP antibody (Clontech, No. 632375; 1:20,000) or an anti-CHC antibody (Agrisera, AS10 690; 1:2000). Anti-Mouse IgG, HRP-Linked Whole Ab Sheep (GE Healthcare, NA931; 1:5000), and Anti-Rabbit IgG, HRP-Linked Whole Ab Donkey (GE Healthcare, NA934; 1:5000) secondary antibodies were used to detect the primary antibodies with ECL select western blotting detection reagent (RPN2235, GE Healthcare). Densitometry was performed with "Quantification of Gel Bands by an Image J Macro, Band/Peak Quantification Tool[69]". The band intensities of CHC were divided by the band intensity of each corresponding GFP-tag, normalized to the AP1M2-GFP line, and expressed as the relative coimmunoprecipitated CHC.

**Statistics.** Sample sizes were at least 30 TGNs from at least three individual plants for colocalization analysis on a TGN. Statistical analyses were performed with R version 3.3.1. Two-sided Steel-Dwass test was used for multiple comparisons. Two-sided Wilcoxon rank-sum test was used for two-group comparisons. P values <0.01 were considered as statistically significant.

**Reporting summary.** Further information on experimental design is available in the Nature Research Reporting Summary linked to this paper.

## Data availability
The authors declare that all data supporting the findings of this study are available within the article and its supplementary information or are available from the corresponding authors on request. Source data are provided with this paper.

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

## Acknowledgements

This work was supported by Grants-in-Aid for Scientific Research from the Ministry of Education, Culture, Sports, Science, and Technology of Japan (grant numbers: 25221103, 17H06420, and 18H05275 to A.N.; 18H04857 to T. Uemura; 17K07477 and 17H06475 to K.T.), and by the Asahi Glass Foundation to T. Uemura. We thank K. Schumacher (Heidelberg University, Germany), I. Hara-Nishimura (Konan University, Japan), T. Shimada (Kyoto University, Japan), S. Mano (National Institute for Basic Biology, Japan), and T. Demura (Nara Institute of Science and Technology, Japan) for sharing materials. We also thank Ichiro Terashima (The University of Tokyo) for helpful discussions.

## Author contributions

Y.S. and T. Uemura designed the study and carried out all the experiments except for electron microscopy. K.E. generated the construct for expressing ST-iRFP and iRFP-VAMP721. Y.G., M.S., and K.T. performed the sampling for and observation by electron microscopy. J.T., E.I., Y.I., Y.K., K.K., and T. Ueda gave advice on this study. Y.S., T. Uemura, and A.N. interpreted the data and wrote the manuscript.

## Competing interests

The authors declare no competing interests.
