## [Peer Review File · Nature Communications]

REVIEWER COMMENTS

Reviewer #1 (Remarks to the Author):

This is a very interesting manuscript describing the use of super-resolution microscopy to identify discrete zones of the TGN that function in different trafficking pathways. This idea has long been suggested, and there are hints in the literature that these zones may exist, but this article is the first to demonstrate them conclusively and describe them in more detail. I have a few suggestions for improvement of the manuscript.

1. What spatial resolution is achieved in the super-resolution microscopy in Arabidopsis? This is important information for anyone who wants to use a similar approach, and as far as I can see the method is published using other organisms but not plants.
2. The paper begins by showing that the TGN SNARES SYP61 and SYP41 localize to a different TGN region than VHAA1. The region containing SYP61 is later shown to be associated with secretion, but the VHAA1-containing region is not further explored. What is the relationship with the zones identified later in the manuscript? Does VHAA1 reside on the vacuolar trafficking zone, or another zone that has yet to be characterized? Do the authors have a hypothesis as to why VHAA1 is not evenly localized throughout the TGN, given its role in acidification?
3. The fluorescence imaging is all performed with components of the trafficking machinery. To confirm that these really do correspond to trafficking routes, co-localization with cargo proteins would be informative. I'm not sure if this is possible in this system, but the authors have previously done a similar experiment in yeast.
4. I find the IP in figure 4h difficult to follow. Are the left hand panels just westerns of the total input before IP? If so, why is CHC missing from the free-GFP sample? The AP-GFP bands are very difficult to see, I am not confident that these are specific. A better description of the experiment and what is shown in the panels is needed.
5. Figure 7 shows that SYP61 leaves the TGN in putative secretory clusters. Does SYP43 behave in the same way? This would be interesting given that *syp61* and *syp4* mutants have overlapping but not identical phenotypes.

Reviewer #2 (Remarks to the Author):

The manuscript by Shimizu et al. uses state of the art 3D localization and 4D dynamics of Trans-Golgi Network (TGN) localized proteins in Arabidopsis thaliana to identify subdomains or the TGN and their role in post Golgi trafficking. The authors used well known proteins, namely the SNARE VAMP727 and the Adaptor Protein 4, that are involved in trafficking of TGN to the vacuole, or to non vacuolar targeted pathways such as VAMP721 along with AP1 to demonstrate the presence of TGN sub compartments and their corresponding routes.

The elegant application of their imaging methodology corroborated earlier studies demonstrating that distinct compartments exist in the TGN thereby sorting cargo to different destinations. The current study provides an excellent visual demonstration of endomembrane dynamics showing that two adaptor proteins AP-1 and AP-4, function in two distinct sorting events in the TGN for secretory and vacuolar trafficking, respectively. Along with the adaptor proteins similarly the dynamics of the two separate SNARES provide an elegant documentation of two sorting zones at the TGN. Such information with the resolution presented in the current study can be used to inform future studies in post Golgi trafficking in plants.

Further, the maturation of TGN in vesicle clusters is well presented, providing hypothesis for future studies. Although very elegantly demonstrated, the results are confirmatory on the role of TGN

and the two proposed sub regions for either secretory or vacuolar trafficking. A novel aspect of the manuscript is the separation of the "secretory" and "vacuolar" zones based on the presence of clathrin. Such information changes the current notion in the field. However, more evidence is required to support this claim.

Below are some suggestions that can improve the current version of the manuscript.

Major comments:

- 1) The manuscript describes subregions of the TGN for either secretory traffic based on the presence or absence of Clathrin CLC2 subunit. Electron microscopy with immunolocalization is necessary to confirm 1) the structure of vesicles identifiable by clathrin 2) the presented here marker proteins for the different subdomains of the TGN in the context of Clathrin.
- 2) The biochemical evidence presented does not provide a clear answer on the separation of AP1 and AP2 based on clathrin. For example in the IP Clathrin is detected in the AP4 IP. Although the authors provide possible scenarios for these data, an alternative biochemical approach is necessary to provide conclusive evidence for the specificity of AP1 with Clathrin as apposed to AP4.
- 3) The proteins selected in both trafficking pathways are involved vesicle fusion/budding, however true cargo proteins are not presented. Soluble cargo in the vacuole or secreted proteins at the apoplast should be used along with the hereby selected SNARES to show the different sorting zones at the TGN.

Minor comment:

Citations provided already in the introduction demonstrate that TGN consists of sub regions including secretory and vacuolar among others. It is notable that the authors did not include many citations that point to sub regions of the TGN such as tethers and other regulatory proteins.

Reviewer #3 (Remarks to the Author):

The paper deals with a significant problem in the field of membrane trafficking. How are proteins are sorted at the TGN to their final destination? There are several routes these cargoes can be directed to. Here they are using 3D and 4D dynamics of several proteins of *Arabidopsis thaliana* involved in cargo transport from the TGN. The authors have developed multicolor high-speed and high-resolution spinning disc confocal microscopy. They show that different TGN localized proteins such as vATPase and Q-SNARES cosegregate in the TGN. Interestingly, they also show that VAMP721 and AP1 form a "secretion zone" while the R-SNARE VAMP727 and AP4 compose a TGN subregion for vacuolar trafficking. The authors have thereby provided evidence that at least two distinct TGN regions responsible for protein sorting. I think the work has been executed at a very high level and show for the first time that compartmentalized functional "zones" of the TGN in high resolution.

Major points:

- 1) Is there a reason that the authors performed their experiments in plant cells? It would be essential to mention it in the text.
- 2) It would be very helpful for non-plant scientists to see how the proteins are distributed in the TGN, for instance, by showing a marker that depicts the entire Golgi. Maybe a nuclear and a cell wall staining would very helpful for the reader.
- 3) The main question of the work is if a cargo of the suggested pathways/zones is following these routes. To answer this question, the authors would need to include a cargo protein that is sorted via AP1 and one that is destined for the vacuolar compartment.
- 4) The authors have performed valuable control experiments to show that the tagging approach does not influence the localization of the proteins, however, it is not clear if the proteins are

overexpressed and if yes to what extent. Please show or comment on this as it could interfere with protein function. I cannot judge if there is a possibility do perform immunofluorescence staining with specific antibodies in plants.

5) I would suggest that the authors track the budding events shown, for instance, in Figure 5 by plotting the intensity profiles.

Minor points:

- 1) Figures 2 and 3 should be combined in one Figure.
- 2) Label what input is in Figure 4.
- 3) The light green color in Figure 8 is not very well visible.

Reviewer #4 (Remarks to the Author):

In this MS, Shimizu and co-authors described that the plant TGN sub-zones are responsible for distinct cargo sorting destinations by super-resolution confocal live imaging microscopy (SCLIM). The authors developed this advanced microscopic technique with improved resolution and application in plant cells. In particular, marker proteins for post-Golgi membrane trafficking to PM or vacuole via TGN were applied for visualising and defining different TGN zones. In general, this MS contains good quality of images and an interesting hypothesis trying to sort out the spatiotemporal functional regions of TGN, yet further experiments and modifications would improve the presentation and conclusions.

Major comments:

1. The study largely used a single microscopic approach to establish the whole system and prove the hypothesis. The choice and use of proper markers are important for making the final conclusions. For example, the Q-SNARE protein SYP61 was used as a marker for the "whole TGN" based on previous publications from using conventional confocal microscopy. It is a possible that under SCLIM with higher resolution SYP61 could also be sub-regionally distributed at TGN, which would raise question on the current conclusions. Therefore, another TGN marker or non-biased dye (e.g. FM4-64) could be applied for proof of concept.
2. Would it be possible to include two distinct cargos that are involved in the secretion and vacuole traffic respectively, in the SCLIM study to demonstrate the two subregions of TGN?
3. Figure 4h, the only biochemical experiment demonstrating the interaction between AP1 but not AP4 with CHC, is a bit confusing and inconsistent. In the input figure, AP1-GFP, AP4-GFP and CHC in the free GFP group can barely be detected even upon over-exposure. This cannot be considered as a fair control and needs improvement. The author must use equal loading of each lines when using Co-IP to compare the protein interaction. In addition, additional protein-protein interacting assays (e.g. Y2H, BiFC, FRET) would provide more solid data for conclusion.
4. The author claimed that CLC1, CLC2 and CLC3 share same localization, thus AP4 does not colocalize with any CLC. This is in sharp contrast comparing with the findings in mammal. The protoplast results are not convincing because 1) the cells are not in good condition in Supplementary Figure 2; and 2) there are more puncta of CLC2 comparing with CLC1 in panel a.
5. Figure 5: The size of the budding AP1M2 positive vesicles from TGN seems to be around 500nm? However, the typical size of CCVs is around 100nm. TEM analysis is suggested to illustrate the nature of the budding profile due to the resolution limitation.
6. Most of the data are derived from super-resolution imaging. Is there any immuno-EM image to

show the different subregions of TGN? Are they morphologically distinct with each other? A discussion would also be useful.

7. What is the expected resolution of the newly developed SCLIM? Can the conclusion from this study be repeated and confirmed by CLEM? It would be interesting to visualize and re-confirm the sub-regions of TGN under TEM.

Other comments:

1. Page 9 lines 130-133, the authors defined the TGN zones using the segregation of VAMP721 and VAMP727 on the TGN under SCLIM. It is suggested to include a control of conventional CLSM to better compare the advancement of the SCLIM. A more discussion of the definition is also advised.

2. Page 17 line 261, the author mentioned a definition "vesicle clusters". What is this concept and could they do more to prove them?

3. Supplementary Fig 2: please include DIC/Bright field image of the protoplast. Some cells look unhealthy and will be not good for experiment.

4. For Supplementary Fig. 3, the brightness is increased to a level where arrows indicated TGN-localized CLC2-mKO are over-exposure and as huge as 1 μm . The authors could include a side-by-side "non-increased" version like they did for the co-IP results.

5. Why the VAMP721 is not fully colocalize with AP1M in Figure 2B if the author claimed that they localize at the similar secretory-trafficking zone?

6. Why the GI-TGN eventually disappear using live-cell imaging? Did the GI-TGN fuse with PM?

7. Methodology part for "Transient assays", the content is more like "cloning technique".

8. All the super-resolution data should include a 3D image rather than just showing the two axis.

9. Some of the original work on plant TGN and AP1 are missing from the citations.

We appreciate all reviewers for warm and thoughtful suggestions to our manuscript.

Reviewer #1 (Remarks to the Author)

This is a very interesting manuscript describing the use of super-resolution microscopy to identify discrete zones of the TGN that function in different trafficking pathways. This idea has long been suggested, and there are hints in the literature that these zones may exist, but this article is the first to demonstrate them conclusively and describe them in more detail. I have a few suggestions for improvement of the manuscript.

1. What spatial resolution is achieved in the super-resolution microscopy in Arabidopsis? This is important information for anyone who wants to use a similar approach, and as far as I can see the method is published using other organisms but not plants.

Response: Using an Argo-SIM slide (Argolight, France) as a nanoscale ruler, we routinely check the xy spatial resolution of SCLIM, which is 180 nm, 180 nm, and 240 nm, for green, red, and infrared channels, respectively. We have also confirmed that SCLIM can resolve adjacent actin filaments with a 180 nm spacing in the red channel in Arabidopsis cells. We have added this information in the methods section (pages 29 and 30, lines 442–444).

2. The paper begins by showing that the TGN SNARES SYP61 and SYP41 localize to a different TGN region than VHAA1. The region containing SYP61 is later shown to be associated with secretion, but the VHAA1-containing region is not further explored. What is the relationship with the zones identified later in the manuscript? Does VHAA1 reside on the vacuolar trafficking zone, or another zone that has yet to be characterized? Do the authors have a hypothesis as to why VHAA1 is not evenly localized throughout the TGN, given its role in acidification?

Response: We thank the reviewer for this constructive comment. According to the reviewer's suggestion, we observed the root cells of Arabidopsis expressing VHAA1-GFP, iRFP-VAMP721 and TagRFP-VAMP727. As shown in Supplementary Fig. 1b and c, the VHAA1-GFP signal overlapped with both iRFP-VAMP721 and TagRFP-VAMP727 without residing on specific trafficking zones. The segregation of SYP43 vs VHAA1 was not as evident as to that of AP-1 vs AP-4, with the higher

correlation of SYP43 and VHAa1 than that of AP1 and AP4 (Fig. 1g and Fig. 2c, median of colocalization correlation coefficient = 0.72 and 0.50, respectively). The meaning of segregation of SYP43 and VHAa1 is not further pursued in this study, but we have added some more statements about it in Results and Discussion (page 9, lines 118–122; page 23, lines 333–337).

3. The fluorescence imaging is all performed with components of the trafficking machinery. To confirm that these really do correspond to trafficking routes, co-localization with cargo proteins would be informative. I'm not sure if this is possible in this system, but the authors have previously done a similar experiment in yeast.

Response: Since R-SNAREs, such as VAMP721 and VAMP727, are believed to travel from the donor compartment to the target during traffic, they can be regarded a kind of cargo following the routes, but we agree, analysis of natural cargo will be very informative. However, because the passage of cargo is transient in the trafficking pathway, it is necessary to establish a pulse-chase-type experimental system. Several methods have been proposed and successfully applied in other organisms, such as retention using selective hooks (RUSH) system in mammalian cultured cells (Boncompain et al., 2012, *Nat. Methods*), and temperature-controlled system in budding yeast (Kurokawa et al., 2014, *Nat. Commun.*). Unfortunately, such systems have not been established in plant cells yet. We previously examined plasma membrane proteins, such as PIN2 and BOR1, but we were unable to detect their signals at the TGN due to their constitutive localization to the plasma membrane in the steady state. We are now trying to establish a modified RUSH system to track cargo proteins in plant cells, and with this, we hope to address this issue in future studies.

4. I find the IP in figure 4h difficult to follow. Are the left hand panels just westerns of the total input before IP? If so, why is CHC missing from the free-GFP sample? The AP-GFP bands are very difficult to see, I am not confident that these are specific. A better description of the experiment and what is shown in the panels is needed.

Response: We apologize for confusing the reviewer. As the reviewer pointed out, the left panels were immunoblot images of the total input (protein extracts) before applying for IP. In our original manuscript, we loaded the samples in a way that the band intensities of GFP-tagged proteins were nearly equal. Therefore, CHC signals in the total input panel varied. In the revised manuscript, we

repeated the same experiment by loading equal volumes of total extracts (input) using AP2M-GFP instead of CLC2-GFP as a positive control. AP2M has been reported to interact with clathrin (Park et al., 2013, *PNAS*; Yamaoka et al., 2013, *Plant Cell*). As shown in new Fig. 3i, the expression levels of AP1M2, AP4M and AP2M-GFP were comparable, which are now seen in the total input. With the results obtained, we were able to draw basically the same conclusion, that is, the amount of clathrin interacting with AP-4 is significantly lower than with AP-1.

5. Figure 7 shows that SYP61 leaves the TGN in putative secretory clusters. Does SYP43 behave in the same way? This would be interesting given that *syp61* and *syp4* mutants have overlapping but not identical phenotypes.

Response: As far as we could observe, SYP43 and SYP61 behave very similarly. We have also observed that, like SYP61, SYP43 moves and disappears together with clathrin. More detailed investigation on the relationship between SYP61 and SYP4s will be interesting to understand their possible role difference, but it is a little beyond the scope of the present study. We will need to trace individual transport vesicles after fragmentation in order to dissect the trafficking routes in detail. Tracking of the dynamics of transport vesicles is unfortunately not possible with the current version of SCLIM. In future experiments, we will tackle these problems using the newly developed second-generation SCLIM, which we hope is capable of analyzing behaviors of individual vesicles.

Reviewer #2 (Remarks to the Author)

The manuscript by Shimizu et al. uses state of the art 3D localization and 4D dynamics of Trans-Golgi Network (TGN) localized proteins in *Arabidopsis thaliana* to identify subdomains of the TGN and their role in post Golgi trafficking. The authors used well known proteins, namely the SNARE VAMP727 and the Adaptor Protein 4, that are involved in trafficking of TGN to the vacuole, or to non vacuolar targeted pathways such as VAMP721 along with AP1 to demonstrate the presence of TGN sub compartments and their corresponding routes.

The elegant application of their imaging methodology corroborated earlier studies demonstrating that distinct compartments exist in the TGN thereby sorting cargo to different destinations. The current study provides an excellent visual demonstration of endomembrane dynamics showing that two adaptor proteins AP-1 and AP-4, function in two distinct sorting events in the TGN for secretory and

vacuolar trafficking, respectively. Along with the adaptor proteins similarly the dynamics of the two separate SNARES provide an elegant documentation of two sorting zones at the TGN. Such information with the resolution presented in the current study can be used to inform future studies in post Golgi trafficking in plants.

Further, the maturation of TGN in vesicle clusters is well presented, providing hypothesis for future studies. Although very elegantly demonstrated, the results are confirmatory on the role of TGN and the two proposed sub regions for either secretory or vacuolar trafficking. A novel aspect of the manuscript is the separation of the “secretory” and “vacuolar” zones based on the presence of clathrin. Such information changes the current notion in the field. However, more evidence is required to support this claim.

Below are some suggestions that can improve the current version of the manuscript.

Major comments:

1) The manuscript describes subregions of the TGN for either secretory traffic based on the presence or absence of Clathrin CLC2 subunit. Electron microscopy with immunolocalization is necessary to confirm 1) the structure of vesicles identifiable by clathrin 2) the presented here marker proteins for the different subdomains of the TGN in the context of Clathrin.

Response: We thank the reviewer for insightful comments. We fully agree with the reviewer’s suggestion that electron microscopic data will support our hypothesis. We tried to address this issue by immunogold labeling of AP1M2-GFP and AP4M-GFP. We performed immunoelectron microscopy experiments, as described previously (Toyooka et al., 2009, *Plant Cell*; Yoshimoto et al., 2014, *JCS*), under the following various conditions with 3 antibodies and 3 fixatives. In brief, Arabidopsis root cells (7 DAG) were frozen in a high-pressure freezer EM-ICE (Leica) and substituted/fixed with acetone containing 1% glutaraldehyde (GA), that containing 1% GA and 1% OsO₄, or that containing 0.25% GA and 0.1% uranyl acetate. The fixed samples were then embedded in LR White resin. Ultrathin sections of the sample were labeled with anti-GFP primary antibodies (Invitrogen: 1/200 for 1h, 1/200 for 3h, or 1/50, for 1h; or Abcam: 1/50 for 1h; or Roche: 1/100 for 2h or 1/500 for 1h; or mix of 1/200 Invitrogen and 1/50 Abcam for 1h). The anti-GFP antibodies from Invitrogen and Abcam were evaluated as useful in the previous studies (Invitrogen A-11122 in Hamada et al., 2018, *JCS*; Abcam ab290 in Le bars et al., 2014, *Nat. Commun.*). However, to our disappointment, we were not able to claim confidently that a zone/subdomain of the TGN was specifically immunolabeled, although we sometimes saw images, in which a part of the TGN appeared to have immunogold signals (Fig.

R1). The low reproducibility was most probably due to very small amounts of antigens under our experimental conditions. Because the fixation conditions for immunolabeling are not suitable for clear resolution of ultrastructures, we were not able to locate clathrin coats in these figures. We decide not make conclusions from these immune-EM data this time. As will be mentioned below in our response to Comment 5 of Reviewer #4, we also carried out conventional EM analysis on the ultrastructures around the TGN region and could obtain images showing interesting vesicle/bud profiles (shown as new Fig. 6d, e). We thank the reviewers for pushing us forward to take collaborative efforts on electron microscopy. Regarding the localization study in ultrahigh resolution, we would like to challenge correlative light and electron microscopy in the future.

2) The biochemical evidence presented does not provide a clear answer on the separation of AP1 and AP2 based on clathrin. For example in the IP Clathrin is detected in the AP4 IP. Although the authors provide possible scenarios for these data, an alternative biochemical approach is necessary to provide conclusive evidence for the specificity of AP1 with Clathrin as apposed to AP4.

Response: To clearly address the differential interaction of clathrin with APs, we tried to perform a yeast two-hybrid assay as an alternative approach (it was also suggested by Reviewer #4). As shown in new Fig. 3h, the yeast two-hybrid analysis using the large subunits of the AP complexes and an amino-terminal domain of CHC clearly showed that γ subunits of AP-1 (AP1G1 and AP1G2) interacted with clathrin but neither ϵ nor β subunits of AP-4 did.

3) The proteins selected in both trafficking pathways are involved vesicle fusion/budding, however true cargo proteins are not presented. Soluble cargo in the vacuole or secreted proteins at the apoplast should be used along with the hereby selected SNARES to show the different sorting zones at the TGN.

Response: As described in our response to Comment 3 of Reviewer #1, the establishment of an experimental system to allow tracking of soluble cargo proteins in plant cells is currently not easy. We previously examined behaviors of plasma membrane cargo proteins, but we were unable to detect the signal at the TGN due to their constitutive localization to the plasma membrane. It was also the case for vacuolar or secreted soluble cargo proteins. We are now trying to develop a RUSH system that is applicable to the plant system to trace cargo proteins, hoping to address this issue in the future. Please

also see our response to Reviewer #1.

Minor comment:

Citations provided already in the introduction demonstrate that TGN consists of sub regions including secretory and vacuolar among others. It is notable that the authors did not include many citations that point to sub regions of the TGN such as tethers and other regulatory proteins.

Response: We have now added several related citations appropriately.

Reviewer #3 (Remarks to the Author)

The paper deals with a significant problem in the field of membrane trafficking. How are proteins are sorted at the TGN to their final destination? There are several routes these cargoes can be directed to. Here they are using 3D and 4D dynamics of several proteins of *Arabidopsis thaliana* involved in cargo transport from the TGN. The authors have developed multicolor high-speed and high-resolution spinning disc confocal microscopy. They show that different TGN localized proteins such as vATPase and Q-SNARES cosegregate in the TGN. Interestingly, they also show that VAMP721 and AP1 form a “secretion zone” while the R-SNARE VAMP727 and AP4 compose a TGN subregion for vacuolar trafficking. The authors have thereby provided evidence that at least two distinct TGN regions responsible for protein sorting. I think the work has been executed at a very high level and show for the first time that compartmentalized functional “zones” of the TGN in high resolution.

Major points:

1) Is there a reason that the authors performed their experiments in plant cells? It would be essential to mention it in the text.

Response: The plant system is very advantageous in dissecting Golgi/TGN dynamics, because unlike other systems, plant cells organize simple and stacked Golgi/TGN under normal conditions, which can be observed as discrete ministacks scattering in the cytoplasm. Plant cells were also the first to demonstrate separation of the TGN from the Golgi stack (Uemura et al., 2004, *Cell Struct. Funct.*;

Staelin & Kang, 2008, *Plant Physiol.*; Viotti et al., 2010, *Plant Cell*). We have mentioned the reason in the discussion section (pages 20 and 21, lines 295–309).

2) It would be very helpful for non-plant scientists to see how the proteins are distributed in the TGN, for instance, by showing a marker that depicts the entire Golgi. Maybe a nuclear and a cell wall staining would very helpful for the reader.

Response: We understand the reviewer's point; however, unfortunately, the entire Golgi marker does not exist in the plant system, and nuclear staining is complicated in our SCLIM spectroscopic system. To define the protein locations with high precision, we are employing high-magnification observation (267x), which is one of the key factors to achieve super-resolution. Observations need to be restricted to the thin cytoplasmic space between the plasma membrane and the large central vacuole in plant root cells. Therefore, the whole plant root cell cannot come into a field of view of cameras. We showed cell boundaries in several figures of the revised manuscript to help understand how cells are aligned.

3) The main question of the work is if a cargo of the suggested pathways/zones is following these routes. To answer this question, the authors would need to include a cargo protein that is sorted via AP1 and one that is destined for the vacuolar compartment.

Response: As described in our responses to Comment 3 of Reviewer #1 and Comment 3 of Reviewer #2, the experimental system to chase cargo proteins in plant cells is unfortunately not available at present. We are now trying to develop a RUSH system in plant cells to track cargo proteins and hope to address this issue in future studies. Please also see our responses to Reviewer #1 and Reviewer #2.

4) The authors have performed valuable control experiments to show that the tagging approach does not influence the localization of the proteins, however, it is not clear if the proteins are overexpressed and if yes to what extent. Please show or comment on this as it could interfere with protein function. I cannot judge if there is a possibility do perform immunofluorescence staining with specific antibodies in plants.

Response: All fluorescent protein-tagged proteins of interest, except for *trans*-Golgi marker (syalyl transferase; ST), were expressed under the control of their own native promoters, not leading to overexpression. Therefore, we believe that our tagging approach does not much influence the physiological states of the plant cells. Indeed, we have shown complementation of Arabidopsis knockout mutants by XFP-tagged protein in some cases (e.g. VAMP727 in Ebine et al., 2008, *Plant cell*; SYP4s in Uemura et al., 2012, *PNAS*). Immunofluorescence staining is possible, but as only a few antibodies are commercially available for the plant system, we need to begin with raising antibodies.

5) I would suggest that the authors track the budding events shown, for instance, in Figure 5 by plotting the intensity profiles.

Response: As the reviewer suggested, we have performed time-course analyses of intensity profiles.

Minor points:

1) Figures 2 and 3 should be combined in one Figure.

Response: As the reviewer suggested, we have combined previous Figures 2 and 3 as new Figure 2.

2) Label what input is in Figure 4.

Response: The input was protein extracts before IP. As described in our response to Comment 4 of Reviewer #1, we repeated Co-IP experiments with some modifications, and replaced corresponding panels in the revised manuscript (Fig. 3i, j). We hope the revised version is much clearer.

3) The light green color in Figure 8 is not very well visible.

Response: We have changed the color to look more vivid.

Reviewer #4 (Remarks to the Author)

In this MS, Shimizu and co-authors described that the plant TGN sub-zones are responsible for distinct cargo sorting destinations by super-resolution confocal live imaging microscopy (SCLIM). The authors developed this advanced microscopic technique with improved resolution and application in plant cells. In particular, marker proteins for post-Golgi membrane trafficking to PM or vacuole via TGN were applied for visualising and defining different TGN zones. In general, this MS contains good quality of images and an interesting hypothesis trying to sort out the spatiotemporal functional regions of TGN, yet further experiments and modifications would improve the presentation and conclusions.

Major comments:

1. The study largely used a single microscopic approach to establish the whole system and prove the hypothesis. The choice and use of proper markers are important for making the final conclusions. For example, the Q-SNARE protein SYP61 was used as a marker for the “whole TGN” based on previous publications from using conventional confocal microscopy. It is possible that under SCLIM with higher resolution SYP61 could also be sub-regionally distributed at TGN, which would raise question on the current conclusions. Therefore, another TGN marker or non-biased dye (e.g. FM4-64) could be applied for proof of concept.

Response: We appreciate the reviewer’s suggestion. We tried to visualize the whole TGN structure by staining the cells with FM4-64 or FM5-95 and observe under SCLIM. However, we were not able find an appropriate experimental condition for simultaneous visualization of FM dyes with two other markers by SCLIM, since the FM fluorescence signals largely overlapped with the mRFP channel. Instead, we performed multi-spectral imaging by linear-unmixing using Carl Zeiss LSM780 equipped with a high-resolution objective lens. As shown in Supplementary Fig. 1f–h, the observation of FM4-64, AP1M2-mRFP, and AP4M-GFP showed that the TGN labeled with FM4-64 certainly had both the secretory-trafficking zone and the vacuolar-trafficking zone.

2. Would it be possible to include two distinct cargos that are involved in the secretion and vacuole

traffic respectively, in the SCLIM study to demonstrate the two subregions of TGN?

Response: We understand the reviewer's concern. It was asked by all reviewers. We agree it is a very important point. However, as described in our responses to other reviewers (i.e. Comment 3 of Reviewer #1, Comment 3 of Reviewer #2 and Comment 3 of Reviewer #3), the experimental system that allows to chase cargo proteins has not been established in plant cells yet. We are now trying to develop a RUSH system that is applicable for plant system to track cargo proteins and hope to address this issue in future studies. Please also see our response to Reviewer #1.

3. Figure 4h, the only biochemical experiment demonstrating the interaction between AP1 but not AP4 with CHC, is a bit confusing and inconsistent. In the input figure, AP1-GFP, AP4-GFP and CHC in the free GFP group can barely be detected even upon over-exposure. This cannot be considered as a fair control and needs improvement. The author must use equal loading of each lines when using Co-IP to compare the protein interaction. In addition, additional protein-protein interacting assays (e.g. Y2H, BiFC, FRET) would provide more solid data for conclusion.

Response: We thank the reviewer for the suggestion. As described in our response to Comment 4 of Reviewer #1, we repeated the coIP experiment with some modifications according to this reviewer's comment and replaced the data in the revised manuscript (Fig. 3i, j). We have also performed yeast-two hybrid assay to test for the interactions between APs and clathrin as another means (Fig. 3h). Please also refer to our response to Comment 2 of Reviewer #2.

4. The author claimed that CLC1, CLC2 and CLC3 share same localization, thus AP4 does not colocalize with any CLC. This is in sharp contrast comparing with the findings in mammal. The protoplast results are not convincing because 1) the cells are not in good condition in Supplementary Figure 2; and 2) there are more puncta of CLC2 comparing with CLC1 in panel a.

Response: We repeated the transient expression experiment carefully. We could reproduce similar results with a better quality, which are now shown in Supplementary Fig. 2e-f. In the revised manuscript, we also performed whole-mount immunofluorescence staining of CHCs to reveal endogenous clathrin localization. As shown in Supplementary Fig. 2a-d, endogenous CHCs

colocalized well with CLC2-GFP and AP1M2-GFP, but not with AP4M-GFP.

5. Figure 5: The size of the budding AP1M2 positive vesicles from TGN seems to be around 500nm? However, the typical size of CCVs is around 100nm. TEM analysis is suggested to illustrate the nature of the budding profile due to the resolution limitation.

Response: We appreciate the reviewer's suggestion very much. We performed TEM analysis. Supporting our hypothesis, the GI-TGN structures showed profiles of multiple clathrin-coated vesicles/buds (Fig. 6d, e). Previous EM studies have also shown that the TGN, like the late/free-TGN (Staelin & Kang, 2008, *Plant Physiol.*; Kang et al., 2011, *Traffic*) or the immature secretory vesicle clusters (Toyooka et al., 2009, *Plant cell*) is composed of non-clathrin-coated vesicles/buds as well as clathrin-coated vesicles/buds. We have now slightly modified our schematic model in Fig. 7 based on these observations.

6. Most of the data are derived from super-resolution imaging. Is there any immuno-EM image to show the different subregions of TGN? Are they morphologically distinct with each other? A discussion would also be useful.

Response: As described in our responses to Comment 1 of Reviewer #2, we tried immunoelectron microscopy experiments. However, to our disappointment, the signals were not significant enough to confidently claim that a zone/subdomain of the TGN was specifically immunolabeled (Fig. R1). We would like to investigate the structural details of the zones of the TGN, for example by CLEM, in our future studies. Please also refer to our response to Reviewer #2.

7. What is the expected resolution of the newly developed SCLIM? Can the conclusion from this study be repeated and confirmed by CLEM? It would be interesting to visualize and re-confirm the sub-regions of TGN under TEM.

Response: In the present study, we used the first-generation SCLIM (SCLIM1), whose resolution is 180-240 nm as described in our response to Comment 1 of Reviewer #1. The newly developed SCLIM (second-generation SCLIM, SCLIM2) has achieved much higher resolution, ca. 70 nm in 2D (to be

published elsewhere soon), but its development is still in progress. We are certainly planning to observe the TGN dynamics by SCLIM2 in future studies. As mentioned above, we would also like to try CLEM in the future. At present, CLEM analysis of marker proteins is technically very complicated.

Other comments:

1. Page 9 lines 130-133, the authors defined the TGN zones using the segregation of VAMP721 and VAMP727 on the TGN under SCLIM. It is suggested to include a control of conventional CLSM to better compare the advancement of the SCLIM. A more discussion of the definition is also advised.

Response: We used iRFP713 (excitation/emission maxima = 690 nm/713 nm) for 3-color observations. This is a key to perform live imaging for 3 proteins of interest in combination with GFP and RFP by SCLIM to assure appropriate fluorescence wavelength windows. Meanwhile, iRFP signal is difficult to detect by our conventional CLSM. Therefore, we were not able to observe this line by conventional CLSM. We hope to better define the zones of the TGN by SCLIM2 and also CLEM, hopefully, in the future.

2. Page 17 line 261, the author mentioned a definition “vesicle clusters”. What is this concept and could they do more to prove them?

Response: As described in our response to the major comment 5 of this reviewer, we have investigated the ultrastructure of the GI-TGN by TEM. As shown in Fig. 6d and e, GI-TGN shows a profile of multiple clathrin-coated vesicles/buds, and the CCV cluster were observed near the Golgi/TGN. These structures are what we would like to consider as vesicle clusters.

3. Supplementary Fig 2: please include DIC/Bright field image of the protoplast. Some cells look unhealthy and will be not good for experiment.

Response: We have added the DIC images according to the reviewer’s suggestion.

4. For Supplementary Fig. 3, the brightness is increased to a level where arrows indicated TGN-

localized CLC2-mKO are over-exposure and as huge as 1 μm . The authors could include a side-by-side “non-increased” version like they did for the co-IP results.

Response: We have replaced it by a new version containing side-by-side views according to the reviewer’s suggestion.

5. Why the VAMP721 is not fully colocalize with AP1M in Figure 2B if the author claimed that they localize at the similar secretory-trafficking zone?

Response: We think that other cargos, such as VAMP722 and/or non-SNARE cargos, could also occupy the secretory-trafficking zone. The same would be true for coat proteins, for example, AP1M1 (AP1M2 isoform) and other adaptors could also be in the secretory-trafficking zone. Thus, these molecules might randomly mix in the zone, possibly leading to mosaic distribution in the same zone.

6. Why the GI-TGN eventually disappear using live-cell imaging? Did the GI-TGN fuse with PM?

Response: In our observation by SCLIM, the GI-TGN seems to be fragmented eventually. In the later stage, the GI-TGN is probably divided into the small and fast-moving vesicles, making it difficult to chase by microscopic observations even by SCLIM. In future studies, we will investigate the behavior of these individual vesicles by the second-generation SCLIM, SCLIM2, which will enable us to trace individual vesicles.

7. Methodology part for “Transient assays”, the content is more like “cloning technique”.

Response: We have divided the original text in “Transient assays” into “Plasmid construction” and “Fluorescence microscopy and image analyses” sections in the methodology part.

8. All the super-resolution data should include a 3D image rather than just showing the two axis.

Response: All SCLIM images were subjected to 3D rendering using the “3D opacity” function of Volocity software, which includes 3D information, such as perspective. Therefore, the images presented in the manuscript are not projected 2D image data. Scale bars are set on the most distant plane. For better representation, we added other angle images for the localization analyses part (Fig. 1–3). For the dynamics analyses part (Fig. 4-6), we did not add other angle images. In the Arabidopsis root epidermal cell of elongation zone, Golgi/TGNs are scattered in the cytoplasm and move vigorously along x and y axes, and therefore we consider that the top view from the z axis is most informative.

9. Some of the original work on plant TGN and AP1 are missing from the citations.

Response: We have added additional citations on plant TGN and AP1.

Fig. R1 Immunogold labeling of AP1M2-GFP or AP4M-GFP with an anti-GFP antibody. **a, b** Arabidopsis root cells expressing AP1M2-GFP (**a**) or AP4M-GFP (**b**) were fixed by high-pressure freezing/freeze substitution with anhydrous acetone containing 0.25% glutaraldehyde and 0.1% uranyl acetate, and ultrathin sections were prepared. The ultrathin sections were stained with anti-GFP antibodies. Arrows show gold particles. ER, Endoplasmic reticulum; G, Golgi; T, GA-TGN; M, Multivesicular endosome; V, vacuole. Scale bars = 200 nm.

REVIEWERS' COMMENTS

Reviewer #1 (Remarks to the Author):

The authors have addressed the majority of my original comments, although they have been unable to address the request from all reviewers to image cargo proteins en route through the TGN to confirm that the different domains correspond to different trafficking pathways. I understand the difficulties inherent in setting up new techniques in plants and in my opinion, on balance, the work is of sufficient interest that it is appropriate for publication without these additional data.

Reviewer #2 (Remarks to the Author):

The manuscript by Shimizu et al. is a revised version of a previously submitted manuscript using state of the art 4D microscopy to elegantly dissect trafficking pathways at the plant TGN. The main concerns were that soluble cargo was not investigated in the presented study to show differential cargo sorting via the described sub regions of TGN and the corresponding pathways. A second concern was the verification of the structural identities of the proposed TGN sub regions as described in the manuscript by live imaging. The revised manuscript however does not provide experimental information addressing directly these two concerns, although the authors describe the technical difficulties preventing these studies.

The authors present EM images in figs 6d,e with a cluster of TGN vesicles including clathrin coated profiles, as expected, but the presented data do not provide EM evidence of TGN separation corresponding to vacuolar or secretory traffic based on clathrin coat.

My third concern on the biochemical evidence for differential interaction of AP1 and AP2 with clathrin was addressed with data derived from a yeast two hybrid assay.

Reviewer #3 (Remarks to the Author):

The authors have comprehensively addressed my concerns. The paper has improved very much, and I would strongly support its acceptance and publication as soon as possible.

Julia von Blume

Response to Editor

We would ask that you make clear how the observed TGN subregions relate to different trafficking pathways has not yet been confirmed and further work would be needed to establish a relationship.

Response: We have added one sentence for this concern in Page 24.

Response to reviewers

Reviewer #1 (Remarks to the Author):

The authors have addressed the majority of my original comments, although they have been unable to address the request from all reviewers to image cargo proteins en route through the TGN to confirm that the different domains correspond to different trafficking pathways. I understand the difficulties inherent in setting up new techniques in plants and in my opinion, on balance, the work is of sufficient interest that it is appropriate for publication without these additional data.

Response: We appreciate the reviewer for understanding the difficulties in setting up techniques for monitoring cargo proteins in plant cells.

Reviewer #2 (Remarks to the Author):

The manuscript by Shimizu et al. is a revised version of a previously submitted manuscript using state of the art 4D microscopy to elegantly dissect trafficking pathways at the plant TGN. The main concerns were that soluble cargo was not investigated in the presented study to show differential cargo sorting via the described sub regions of TGN and the corresponding pathways. A second concern was the verification of the structural identities of the proposed TGN sub regions as described in the manuscript by live imaging. The revised manuscript however does not provide experimental information addressing directly these two concerns, although the authors describe the technical difficulties preventing these studies. The authors present EM images in figs 6d,e with a cluster of TGN vesicles including clathrin coated profiles, as expected, but the presented data do not provide EM evidence of TGN separation corresponding to vacuolar or secretory traffic based on clathrin coat. My third concern on the biochemical evidence for differential interaction of AP1 and AP2 with clathrin was addressed with data derived from a yeast two hybrid assay.

Response: We have made clear that how the observed TGN zones relate to different trafficking pathways remains to be confirmed directly. (Page xx line xx.)

Reviewer #3 (Remarks to the Author):

The authors have comprehensively addressed my concerns. The paper has improved very much, and I would strongly support its acceptance and publication as soon as possible.

Julia von Blume

Response: We are very happy to see this comment. We sincerely appreciate your support.

We are deeply grateful to all the reviewers for constructive and valuable comments, which have improved our work very much.